# Beyond Attention or Similarity: Maximizing Conditional Diversity for Token Pruning in MLLMs

**Qizhe Zhang**[1][*]  **Mengzhen Liu**[1]  **Lichen Li**[1]  **Ming Lu**[1][†]
**Yuan Zhang**[1]  **Junwen Pan**[2]  **Qi She**[2][†]  **Shanghang Zhang**[1][✉]
[1] State Key Laboratory for Multimedia Information Processing,
School of Computer Science, Peking University  [2] ByteDance
{theia, shanghang}@pku.edu.cn

## Abstract

In multimodal large language models (MLLMs), the length of input visual tokens is often significantly greater than that of their textual counterparts, leading to a high inference cost. Many works aim to address this issue by removing redundant visual tokens. However, current approaches either rely on attention-based pruning, which retains numerous duplicate tokens, or use similarity-based pruning, overlooking the instruction relevance, consequently causing suboptimal performance. In this paper, we go beyond attention or similarity by proposing a novel visual token pruning method named **CDPruner**, which maximizes the conditional diversity of retained tokens. We first define the conditional similarity between visual tokens conditioned on the instruction, and then reformulate the token pruning problem with determinantal point process (DPP) to maximize the conditional diversity of the selected subset. The proposed CDPruner is training-free and model-agnostic, allowing easy application to various MLLMs. Extensive experiments across diverse MLLMs show that CDPruner establishes new state-of-the-art on various vision-language benchmarks. By maximizing conditional diversity through DPP, the selected subset better represents the input images while closely adhering to user instructions, thereby preserving strong performance even with high reduction ratios. When applied to LLaVA, CDPruner reduces FLOPs by 95% and CUDA latency by 78%, while maintaining 94% of the original accuracy. Our code is available at https://github.com/Theia-4869/CDPruner.

## 1 Introduction

Benefiting from the remarkable success of large language models (LLMs) [Touvron et al., 2023a,b, Jiang et al., 2023, Bai et al., 2023, Yang et al., 2024a, Cai et al., 2024b], multimodal large language models (MLLMs) [Liu et al., 2023, 2024a, Wang et al., 2024, Chen et al., 2024d,c, An et al., 2025] have extended their powerful reasoning capabilities to more modalities, such as images or videos. To fully leverage the strengths of LLMs, MLLMs typically encode visual inputs into a form that language models can understand, known as tokens. Within the input sequence, the length of visual tokens often numbers in the hundreds, exceeding their textual counterparts by tens of times. And in video streams [Zhang et al., 2023, Lin et al., 2023, Zhang et al., 2024c] or high-resolution [Liu et al., 2024b, Luo et al., 2024, Guo et al., 2024] scenarios, this number can grow even larger. Since attention-based models [Vaswani et al., 2017] exhibit computational complexity that scales quadratically with token length, an excessive number of visual tokens makes the use of MLLMs costly and impractical for low-latency or resource-constrained applications. [Team et al., 2024, Hu et al., 2024a].

---

[*]Work done during an internship at ByteDance.
[†]Project lead.  [✉]Corresponding author.

39th Conference on Neural Information Processing Systems (NeurIPS 2025).

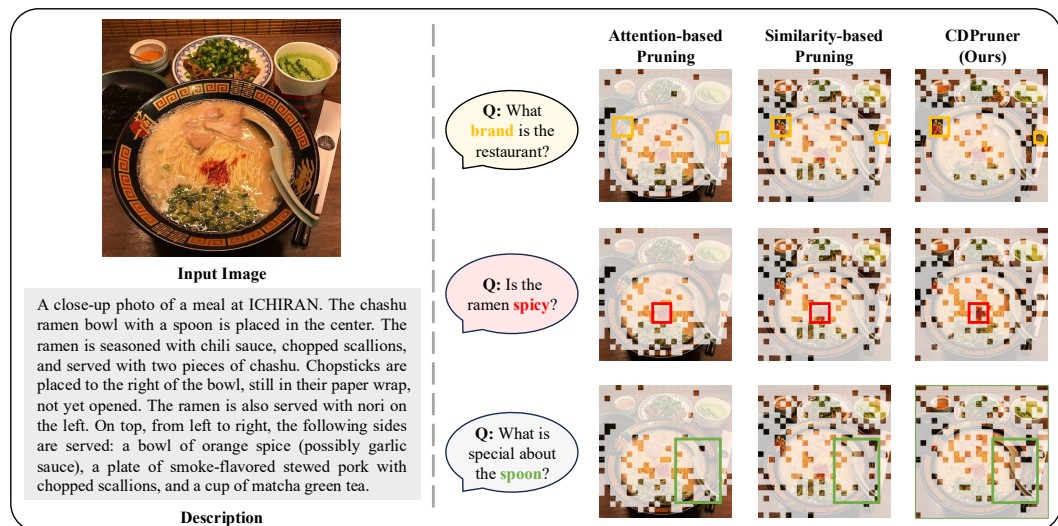

Figure 1: **Comparison of different token pruning methods.** Attention-based methods retain numerous duplicate tokens, failing to achieve effective visual token compression. Similarity-based methods neglect user instructions, always pruning the same tokens and paying insufficient attention to regions most relevant to the question. Our CDPruner considers the conditional diversity of the selected subset, dynamically adjusting pruning according to the user instructions and retaining maximal visual information. In this example, CDPruner successfully preserves tokens related to crucial details, such as the "ICHIRAN" logo on the bowl and chopsticks, the **chili pepper** on the ramen, and the **anti-slip design** on the spoon handle, while both alternative methods fail.

Abundant efforts have been made to reduce the inference cost of MLLMs by pruning visual tokens, and existing methods can be roughly divided into two categories. The first is to identify visual tokens with high attention scores as important and discard those deemed less critical [Chen et al., 2024a, Xing et al., 2024, Zhang et al., 2024b]. The second is to remove redundant parts based on feature similarity between visual tokens [Wen et al., 2025b, Alvar et al., 2025, Jeddi et al., 2025]. As illustrated in Figure 1, both approaches suffer from inherent weaknesses, leading to suboptimal performance after pruning. Attention-based methods only consider the importance of visual tokens, resulting in a large number of duplicate tokens being retained, while similarity-based methods neglect user instructions, failing to achieve dynamic pruning in alignment with the current question.

To address these issues, we propose CDPruner, a plug-and-play method for MLLM inference acceleration by maximizing the conditional diversity of the selected subset. Conditional diversity simultaneously considers feature similarity and instruction relevance, maintaining considerable performance at high reduction ratios without the need for additional training. Specifically, we first calculate pairwise similarity between visual tokens conditioned on their relevance to the input instruction. To obtain the retained tokens, we reformulate the token pruning problem with determinantal point process (DPP), which is widely used for modeling list-wise diversity based on pairwise similarity [Kulesza et al., 2012, Chen et al., 2018, Celis et al., 2018, Li et al., 2024c, Sun et al., 2025]. Direct MAP inference for DPP is NP-hard. To address this, we adopt a greedy algorithm with polynomial-time complexity that guarantees a $(1 - 1/e)$ approximation. By leveraging Cholesky decomposition, the additional latency introduced by solving the DPP remains within the limits required for real-time applications. In practice, the computational complexity can be further reduced through techniques such as sliding-window [Chen et al., 2018] or Markov chain [Kang, 2013] approximations.

As a simple yet effective solution, CDPruner offers several practical advantages. First, in contrast to attention-based methods [Chen et al., 2024a, Xing et al., 2024, Zhang et al., 2024b], CDPruner does not require access to attention scores, which ensures its complete compatibility with efficient attention acceleration implementations [Dao et al., 2022]. Second, CDPruner does not depend on a specific visual encoder or language model, and can be readily implemented across any token-based MLLM [Li et al., 2024a, Bai et al., 2025, Zhu et al., 2025]. Extensive experiments across various MLLMs demonstrate the effectiveness and efficiency of CDPruner. When applied to LLaVA-NeXT-7B, it reduces FLOPs by 95%, CUDA latency by 78%, and GPU memory by 17%, while maintaining 94% of the original performance in a training-free manner.

In summary, the contributions of our work are three-fold:

1. We introduce CDPruner, a plug-and-play and model-agnostic solution for visual token pruning that maximizes conditional diversity.

2. We reformulate the token pruning problem with determinantal point process, which facilitates dynamic pruning by jointly considering feature similarity and instruction relevance.

3. We conduct extensive experiments on various vision-language benchmarks, demonstrating that CDPruner consistently achieves state-of-the-art across different reduction ratios.

## 2 Related work

**Multimodal large language models.** The remarkable achievements of large language models (LLMs) [Touvron et al., 2023a,b, Jiang et al., 2023, Bai et al., 2023, Yang et al., 2024a, Cai et al., 2024b] have lead to a growing trend of extending their powerful reasoning capabilities to other modalities, eventually forming multimodal large language models (MLLMs) [Liu et al., 2023, Li et al., 2024a, Wang et al., 2024, Bai et al., 2025, Chen et al., 2024c, Zhu et al., 2025, Liu et al., 2024c]. These models typically encode visual inputs as tokens to fully leverage the capabilities of LLMs. However, the sparsity of visual signals results in a significantly larger number of visual tokens compared to their textual counterparts. For example, LLaVA-1.5 [Liu et al., 2024a] converts a 336×336 image into 576 tokens, while its high-resolution variant, LLaVA-NeXT [Liu et al., 2024b], generates 2,880 tokens from an image with twice the resolution. In video understanding scenarios, LongVA [Zhang et al., 2024a] transforms 2,000 frames into over 200K visual tokens, and LongVILA [Chen et al., 2024b] can even handle up to 6,000 frames and produce an ultra-long input sequence of over 1M visual tokens, leading to enormous computational overhead. Therefore, achieving more efficient inference for MLLMs is becoming increasingly critical.

**Visual token reduction.** Reducing the number of input visual tokens is an effective way for MLLM inference acceleration. Some works attempt to compress visual tokens via vision-text pre-fusion [Li et al., 2024d, Hu et al., 2024b, Cai et al., 2024a, Zhang et al., 2025b], but these approaches require architectural modifications and additional training, thereby increasing computational costs. Other works adopt a training-free approach by removing redundant visual tokens during inference [Liu et al., 2024d, Yang et al., 2025, Cao et al., 2025, Ma et al., 2025, Zhang et al., 2025c], known as token pruning. These methods can be broadly categorized into three groups.

The first group leverages text-visual attentions within the language model to assess the importance of visual tokens [Chen et al., 2024a, Ye et al., 2025, Xing et al., 2024, Zhang et al., 2024b]. However, as pointed out by Zhang et al. [2025a] and Wen et al. [2025a], such methods suffer from attention shift, which compromises pruning accuracy. Moreover, the reliance on attention scores makes them incompatible with efficient attention implementations like FlashAttention [Dao et al., 2022]. The second group avoids these issues by pruning before the language model [Shang et al., 2024, Yang et al., 2024b, Song et al., 2024, Zhang et al., 2025a]. Nonetheless, these methods rely on specific visual encoder architectures and thus cannot be applied across different MLLMs. The third group directly prunes tokens based on feature similarity among visual tokens [Wen et al., 2025b, Alvar et al., 2025, Jeddi et al., 2025]. However, like the second group, they fail to consider the relevance between visual tokens and user instructions during pruning, leading to suboptimal performance. In this work, our CDPruner addresses all these challenges by jointly modeling feature similarity and instruction relevance through DPP, thereby ensuring both the diversity and quality of the retained token subset.

**Determinantal point process.** Determinantal Point Process (DPP) was first introduced to describe the distribution of fermion systems in thermal equilibrium [Macchi, 1975], where no two fermions can occupy the same quantum state, resulting in an "anti-bunching" effect that can be interpreted as diversity. Later, DPPs have been widely adopted in list-wise diversity modeling across various domains [Chen et al., 2018, Celis et al., 2018, Li et al., 2024c, Sun et al., 2025]. Unlike Max-Min Diversity Problem (MMDP) [Porumbel et al., 2011], which also aims to maximize diversity, DPP emphasizes global diversity and typically yields more balanced and representative subset selections [Kulesza et al., 2012]. Traditional DPP focuses solely on feature similarity among samples. In this work, we extend this formulation by incorporating instruction relevance as a condition, enabling a unified consideration for superior visual token pruning performance in MLLMs.

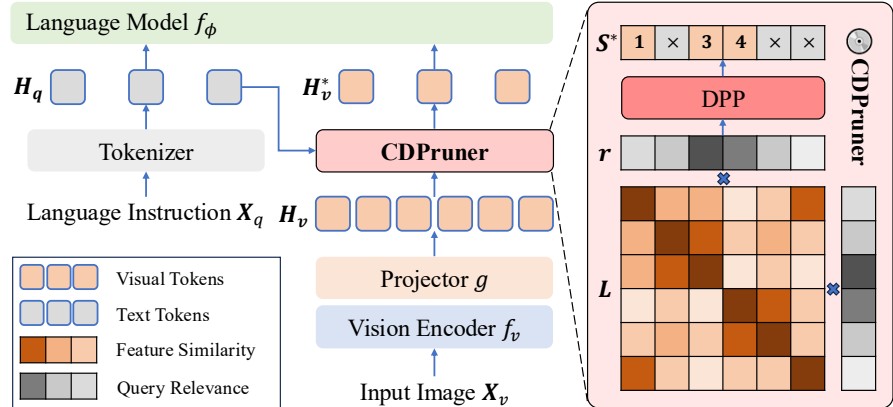

Figure 2: **Overview of CDPruner.** We first calculate the similarity between visual tokens conditioned on their relevance to the current instruction. Then, CDPruner uses a DPP to select the subset to keep. As a training-free and model-agnostic method, it ensures both the diversity and quality of the selected token subset, significantly reducing computational cost while maintaining considerable performance.

## 3 Method

In this section, we first review visual token pruning in MLLMs in Section 3.1. Then, we model the feature similarity among visual tokens and their relevance to user instructions in Section 3.2 and Section 3.3. Finally, we present our CDPruner in Section 3.4, which maximizes the conditional diversity to obtain the optimal token subset. The overall design of CDPruner is shown in Figure 2.

### 3.1 Visual token pruning

Existing MLLMs [Liu et al., 2024a, Wang et al., 2024, Chen et al., 2024c] typically consist of three core components: a vision encoder $f_v$, a multimodal projector $g$, and an LLM $f_\phi$. The vision encoder encodes the input image $X_v$ into a sequence of visual tokens $\boldsymbol{H}_v = g(f_v(X_v)) \in \mathbb{R}^{n \times d}$, whose length is significantly greater than that of their textual counterparts $\boldsymbol{H}_q$. Visual token pruning aims to reduce the inference cost of MLLMs by decreasing the number of visual tokens:

$$\tilde{\boldsymbol{H}}_v^* = \underset{\tilde{\boldsymbol{H}}_v \subseteq \boldsymbol{H}_v, |\tilde{\boldsymbol{H}}_v| = m}{\arg \min} \mathcal{L}\left(f_\phi([\tilde{\boldsymbol{H}}_v; \boldsymbol{H}_q]), f_\phi([\boldsymbol{H}_v; \boldsymbol{H}_q])\right). \tag{1}$$

Here, $\mathcal{L}$ measures the discrepancy between the model outputs before and after visual token pruning, and $m$ is the number of visual tokens retained ($m < n$). Previous methods mainly rely on attention scores for pruning [Chen et al., 2024a, Xing et al., 2024, Zhang et al., 2024b, Shang et al., 2024, Yang et al., 2024b], which often leads to significant redundancy. Alvar et al. [2025] formulates the subset selection problem as a Max-Min Diversity Problem (MMDP) [Porumbel et al., 2011], but this approach overly focuses on extreme cases while neglecting global diversity.

### 3.2 DPP with token similarity

DPP was initially introduced to model fermion repulsion in quantum physics [Macchi, 1975], and has been widely applied in list-wise diversity modeling [Chen et al., 2018, Celis et al., 2018, Sun et al., 2025]. Formally, a DPP $\mathcal{P}$ on a discrete set $Z = \{1, 2, \ldots, n\}$ is a probability measure defined on the power set $2^Z$. When $\mathcal{P}$ gives nonzero probability to the empty set, there exists a positive semi-definite (PSD) kernel matrix $\boldsymbol{L} \in \mathbb{R}^{n \times n}$ indexed by elements of $Z$, such that for every subset $S \subseteq Z$, the probability of sampling $S$ is:

$$\mathcal{P}(S) = \frac{\det(\boldsymbol{L}_S)}{\det(\boldsymbol{L} + \boldsymbol{I})} \propto \det(\boldsymbol{L}_S), \tag{2}$$

where $\boldsymbol{L}_S$ is the principal submatrix of $\boldsymbol{L}$ corresponding to the subset $S$.

In the context of token pruning, we leverage DPP to model the diversity of the retained visual token subset. Given a sequence of visual tokens $\boldsymbol{H}_v$, the kernel matrix $\boldsymbol{L}$ is defined by the pairwise cosine

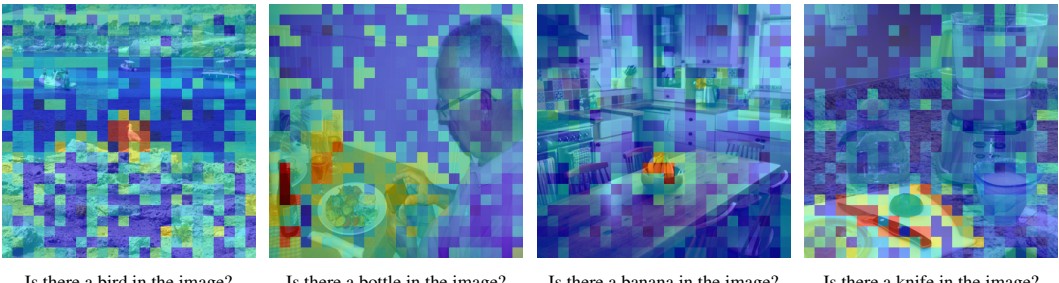

| Is there a bird in the image? | Is there a bottle in the image? | Is there a banana in the image? | Is there a knife in the image? |

Figure 3: **Visualization of relevance scores.** We compute the relevance scores for several samples from the POPE benchmark using LLaVA-1.5-7B, with the instruction following the template: "Is there a {object} in the image?" **Red** indicates high relevance, while **blue** indicates low relevance.

similarity of visual features:

$$L_{ij} = \frac{\boldsymbol{H}_v^i \cdot \boldsymbol{H}_v^j}{\|\boldsymbol{H}_v^i\| \cdot \|\boldsymbol{H}_v^j\|}. \tag{3}$$

According to the DPP sampling process, the optimal subset $\tilde{\boldsymbol{H}}_v^*$ is given by:

$$S^* = \underset{S \subseteq Z, |S|=m}{\arg\max} \, \det\left(\boldsymbol{L}_S\right), \quad \tilde{\boldsymbol{H}}_v^* = \left\{\boldsymbol{H}_v^i \mid i \in S^*\right\}. \tag{4}$$

### 3.3 Instruction relevance

The above only considers the feature similarity among visual tokens, resulting in the same pruning result regardless of user instructions. We further introduce instruction relevance as a condition to achieve dynamic pruning. Given the visual embeddings $\boldsymbol{H}_v \in \mathbb{R}^{n \times d}$ extracted from the input image and the text embeddings $\bar{\boldsymbol{H}}_q \in \mathbb{R}^d$ derived from the user instruction, we calculate the cosine similarity to measure the relevance $\boldsymbol{r} \in \mathbb{R}^n$ between each visual token and the instruction:

$$r_i = \frac{\boldsymbol{H}_v^i \cdot \bar{\boldsymbol{H}}_q}{\|\boldsymbol{H}_v^i\| \cdot \|\bar{\boldsymbol{H}}_q\|}. \tag{5}$$

For MLLMs [Liu et al., 2023, 2024b, Li et al., 2024a] that employ visual encoders paired with corresponding text encoders (e.g., CLIP [Radford et al., 2021]), we use features extracted from both as visual and text embeddings, respectively. For MLLMs [Bai et al., 2025, Zhu et al., 2025] only contain dedicated visual encoders, we instead use the output of the multimodal projector as the visual embeddings, and take the average of all token embeddings corresponding to the instruction from the language model as the text embedding. For simplicity, we denote the visual and text embeddings obtained through both ways as $\boldsymbol{H}_v$ and $\bar{\boldsymbol{H}}_q$. Figure 3 shows the relevance scores derived through the LLaVA-1.5-7B [Liu et al., 2024a] for several samples from the POPE benchmark [Li et al., 2023].

Furthermore, we apply min-max normalization to the obtained relevance scores to ensure the values are within the range of 0 to 1:

$$\tilde{\boldsymbol{r}} = \frac{\boldsymbol{r} - \min(\boldsymbol{r})}{\max(\boldsymbol{r}) - \min(\boldsymbol{r})}. \tag{6}$$

### 3.4 CDPruner

Finally, we integrate feature similarity and instruction relevance for visual token pruning, leading to our proposed **CDPruner**, as shown in Figure 2. Specifically, we modulate the original kernel matrix with the relevance scores to obtain a new conditional kernel matrix:

$$\tilde{\boldsymbol{L}} = \operatorname{diag}\left(\tilde{\boldsymbol{r}}\right) \cdot \boldsymbol{L} \cdot \operatorname{diag}\left(\tilde{\boldsymbol{r}}\right). \tag{7}$$

The updated log-probability of the subset $S$ for DPP is:

$$\log \det\left(\tilde{\boldsymbol{L}}_S\right) = \sum_{i \in S} \log\left(\tilde{\boldsymbol{r}}_i^2\right) + \log \det\left(\boldsymbol{L}_S\right). \tag{8}$$

Table 1: **Performance comparison of different pruning methods on LLaVA-1.5-7B.** Here, **Acc.** denotes the average performance across 10 benchmarks, **Rel.** represents the average percentage of performance maintained. Attention-based methods are shown with red background, attention&similarity-based methods with green background, and similarity-based methods with blue background.

| Method | VQA$^{V2}$ | GQA | VizWiz | SQA$^{IMG}$ | VQA$^{Text}$ | POPE | MME | MMB$^{EN}$ | MMB$^{CN}$ | MMVet | Acc. | Rel. |
|---|---|---|---|---|---|---|---|---|---|---|---|---|
| *Upper Bound, All 576 Tokens (100%)* | | | | | | | | | | | | |
| LLaVA-1.5-7B | 78.5 | 61.9 | 50.1 | 69.5 | 58.2 | 85.9 | 1506.5 | 64.7 | 58.1 | 31.3 | 63.4 | 100.0% |
| *Retain 128 Tokens (↓ 77.8%)* | | | | | | | | | | | | |
| FastV(ECCV24) | 71.0 | 54.0 | 51.9 | 69.2 | 56.4 | 68.2 | 1368.9 | 63.0 | 55.9 | 27.0 | 58.5 | 92.8% |
| PDrop(CVPR25) | 74.3 | 57.1 | 49.4 | 70.1 | 56.7 | 77.5 | 1444.1 | 62.3 | 55.3 | 27.6 | 60.3 | 95.0% |
| SparseVLM(ICML25) | 75.1 | 57.3 | 49.7 | 69.0 | 56.3 | 83.1 | 1399.3 | 62.6 | 56.9 | 29.7 | 61.0 | 96.3% |
| PruMerge+(2024.05) | 75.0 | 58.2 | 53.7 | 69.1 | 54.0 | 83.1 | 1408.1 | 61.8 | 55.8 | 30.4 | 61.2 | 96.8% |
| TRIM(COLING25) | 75.4 | 58.4 | 51.6 | 68.6 | 52.2 | 85.3 | 1413.4 | 63.0 | 52.3 | 29.9 | 60.7 | 95.8% |
| VisionZip(CVPR25) | 75.6 | 57.6 | 51.6 | 68.7 | 56.9 | 83.3 | 1436.9 | 62.1 | 57.0 | 31.6 | 61.6 | 97.6% |
| DART(2025.02) | 74.7 | 57.9 | 52.8 | 69.1 | 56.3 | 80.4 | 1408.7 | 60.7 | 57.3 | 30.9 | 61.1 | 96.9% |
| DivPrune(CVPR25) | 76.0 | 59.4 | 52.8 | 68.6 | 55.9 | 87.0 | 1405.1 | 61.5 | 54.8 | 30.6 | 61.7 | 97.5% |
| **CDPruner(Ours)** | 76.6 | 59.9 | 52.8 | 69.0 | 56.2 | 87.7 | 1431.4 | 63.1 | 55.0 | 32.8 | **62.5** | **99.0%** |
| *Retain 64 Tokens (↓ 88.9%)* | | | | | | | | | | | | |
| FastV(ECCV24) | 55.9 | 46.0 | 49.1 | 70.1 | 51.6 | 35.5 | 973.5 | 50.1 | 42.1 | 18.9 | 46.8 | 74.9% |
| PDrop(CVPR25) | 56.3 | 46.1 | 46.3 | 68.8 | 49.2 | 40.8 | 982.2 | 48.0 | 36.6 | 17.7 | 45.9 | 72.9% |
| SparseVLM(ICML25) | 66.9 | 52.0 | 49.4 | 69.2 | 52.1 | 69.7 | 1190.4 | 58.3 | 49.6 | 24.4 | 55.1 | 87.1% |
| PruMerge+(2024.05) | 71.3 | 55.4 | 53.7 | 69.5 | 52.0 | 75.7 | 1316.8 | 59.6 | 52.1 | 28.0 | 58.3 | 92.4% |
| TRIM(COLING25) | 72.4 | 56.6 | 51.1 | 69.0 | 49.7 | 85.9 | 1350.9 | 60.9 | 48.2 | 24.8 | 58.6 | 91.6% |
| VisionZip(CVPR25) | 72.4 | 55.1 | 52.9 | 69.0 | 55.5 | 77.0 | 1365.2 | 60.1 | 55.4 | 29.4 | 59.5 | 94.4% |
| DART(2025.02) | 71.3 | 54.7 | 53.5 | 69.3 | 54.7 | 73.8 | 1365.1 | 59.5 | 54.0 | 26.5 | 58.6 | 92.6% |
| DivPrune(CVPR25) | 74.1 | 57.5 | 53.6 | 68.0 | 54.5 | 85.5 | 1334.7 | 60.1 | 52.3 | 28.1 | 60.0 | 94.7% |
| **CDPruner(Ours)** | 75.4 | 58.6 | 53.4 | 68.1 | 55.3 | 87.5 | 1415.1 | 61.1 | 53.2 | 30.5 | **61.4** | **97.0%** |
| *Retain 32 Tokens (↓ 94.4%)* | | | | | | | | | | | | |
| PruMerge+(2024.05) | 65.6 | 52.9 | 53.5 | 67.9 | 49.2 | 66.7 | 1236.6 | 55.1 | 45.9 | 24.7 | 54.3 | 86.1% |
| TRIM(COLING25) | 68.6 | 54.5 | 50.7 | 68.1 | 47.6 | 84.9 | 1251.8 | 57.7 | 40.1 | 20.5 | 55.5 | 86.2% |
| VisionZip(CVPR25) | 67.1 | 51.8 | 52.4 | 69.1 | 53.1 | 69.4 | 1251.2 | 57.0 | 50.3 | 25.3 | 55.8 | 88.4% |
| DART(2025.02) | 67.1 | 52.9 | 52.5 | 69.3 | 52.2 | 69.1 | 1273.3 | 58.5 | 50.0 | 25.0 | 56.0 | 88.6% |
| DivPrune(CVPR25) | 71.2 | 54.9 | 53.3 | 68.6 | 52.9 | 81.5 | 1284.9 | 57.6 | 49.1 | 26.3 | 58.0 | 91.3% |
| **CDPruner(Ours)** | 73.6 | 57.0 | 53.1 | 69.5 | 53.2 | 87.9 | 1373.0 | 59.6 | 49.6 | 27.8 | **60.0** | **94.3%** |

We then obtain the optimal subset via MAP inference. Although MAP inference for DPP is NP-hard, there exists a greedy algorithm with polynomial-time complexity that guarantees a $(1 - 1/e)$ approximation [Chen et al., 2018]. By using Cholesky decomposition, the overall time complexity can be reduced to $\mathcal{O}(nm^2)$. The additional latency is negligible when $m \ll n$, with less than 10ms per sample. The pseudocode for algorithm implementation is provided in the technical appendix.

# 4 Experiments

## 4.1 Experimental setup

**Model architectures.** We apply CDPruner to various MLLM architectures, including the LLaVA series such as LLaVA-1.5 [Liu et al., 2024a] for image understanding, LLaVA-NeXT [Liu et al., 2024b] for high-resolution inputs, and LLaVA-Video [Zhang et al., 2024c] for video understanding, as well as the current state-of-the-art open-source model Qwen2.5-VL [Bai et al., 2025]. Additional results on more model architectures are provided in the technical appendix.

**Evaluation benchmarks.** We evaluate our method on 14 image-based multimodal benchmarks, including 10 general VQA tasks such as VQAv2 [Goyal et al., 2017], GQA [Hudson and Manning, 2019], VizWiz [Gurari et al., 2018], ScienceQA-IMG [Lu et al., 2022], HallBench [Guan et al., 2024], POPE [Li et al., 2023], MME [Fu et al., 2024a], MMBench [Liu et al., 2025], MMBench-CN [Liu et al., 2025] and MM-Vet [Yu et al., 2023], 4 text-oriented VQA tasks such as TextVQA [Singh et al., 2019], ChartQA [Masry et al., 2022], AI2D [Kembhavi et al., 2016] and OCRBench [Liu et al., 2024e], and 1 multi-turn dialog task MMDU [Liu et al., 2024f]. We also conduct experiments on 4 widely-used video understanding benchmarks, including MLVU [Zhou et al., 2024], MVBench [Li et al., 2024b], LongVideoBench [Wu et al., 2024] and Video-MME [Fu et al., 2024b]. All experiments on these benchmarks follow the default settings and evaluation metrics. Detailed descriptions of each task are provided in the technical appendix.

**Comparison methods.** We choose several recent works of different types as comparsion methods, including attention-based methods like FastV [Chen et al., 2024a], PyramidDrop [Xing et al., 2024] and SparseVLM [Zhang et al., 2024b], attention&similarity-based methods like LLaVA-Prumerge [Shang et al., 2024], TRIM [Song et al., 2024] and VisionZip [Yang et al., 2024b], as well as similarity-based methods like DART [Wen et al., 2025b] and DivPrune [Alvar et al., 2025].

Table 2: **Performance comparison of different pruning methods on LLaVA-NeXT-7B. Acc.** denotes the average performance across 10 benchmarks, **Rel.** represents the average percentage of performance maintained. Attention-based methods are shown with red background, attention&similarity-based methods with green background, and similarity-based methods with blue background.

| Method | VQA$^{V2}$ | GQA | VizWiz | SQA$^{IMG}$ | VQA$^{Text}$ | POPE | MME | MMB$^{EN}$ | MMB$^{CN}$ | MMVet | Acc. | Rel. |
|---|---|---|---|---|---|---|---|---|---|---|---|---|
| *Upper Bound, All 2880 Tokens (**100%**)* | | | | | | | | | | | | |
| LLaVA-NeXT-7B | 81.3 | 62.5 | 55.2 | 67.5 | 60.3 | 86.8 | 1511.8 | 65.8 | 57.3 | 40.0 | 65.2 | 100.0% |
| *Retain 640 Tokens (↓ 77.8%)* | | | | | | | | | | | | |
| FastV (ECCV24) | 77.0 | 58.9 | 53.9 | 67.4 | 58.1 | 79.5 | 1412.6 | 63.1 | 53.5 | 39.5 | 62.2 | 95.6% |
| PDrop (CVPR25) | 79.1 | 60.0 | 53.8 | 66.7 | 57.8 | 83.8 | 1475.9 | 64.1 | 55.2 | 36.7 | 63.1 | 96.5% |
| SparseVLM (ICML25) | 79.2 | 61.2 | 53.6 | 67.6 | 59.7 | 85.3 | 1456.8 | 65.9 | 58.6 | 36.1 | 64.0 | 97.9% |
| PruMerge+ (2024.05) | 78.2 | 60.8 | 57.9 | 67.8 | 54.9 | 85.3 | 1480.2 | 64.6 | 57.3 | 32.7 | 63.4 | 96.6% |
| TRIM (COLING25) | 78.3 | 62.1 | 54.8 | 66.9 | 54.8 | 86.9 | 1471.8 | 66.8 | 55.8 | 37.8 | 63.8 | 97.6% |
| VisionZip (CVPR25) | 79.1 | 61.2 | 57.1 | 68.1 | 59.9 | 86.0 | 1493.4 | 65.8 | 58.1 | 38.9 | 64.9 | 99.5% |
| DART (2025.02) | 78.3 | 61.3 | 57.0 | 68.2 | 59.5 | 85.0 | 1450.2 | 64.9 | 57.1 | 36.9 | 64.1 | 98.2% |
| DivPrune (CVPR25) | 79.3 | 61.9 | 55.7 | 67.8 | 57.0 | 86.9 | 1469.7 | 65.8 | 57.3 | 38.0 | 64.3 | 98.5% |
| **CDPruner (Ours)** | 79.9 | 62.6 | 55.6 | 67.9 | 58.4 | 87.3 | 1474.5 | 66.3 | 57.5 | 41.9 | **65.1** | **100.1%** |
| *Retain 320 Tokens (↓ 88.9%)* | | | | | | | | | | | | |
| FastV (ECCV24) | 61.5 | 49.8 | 51.3 | 66.6 | 52.2 | 49.5 | 1099.0 | 53.4 | 42.5 | 20.0 | 50.2 | 76.9% |
| PDrop (CVPR25) | 66.8 | 50.4 | 49.7 | 66.7 | 49.0 | 60.8 | 1171.5 | 55.5 | 44.7 | 24.0 | 52.6 | 80.3% |
| SparseVLM (ICML25) | 74.6 | 57.9 | 54.2 | 67.2 | 56.5 | 76.9 | 1386.1 | 63.1 | 56.7 | 32.8 | 60.9 | 93.3% |
| PruMerge+ (2024.05) | 75.3 | 58.8 | 57.7 | 68.1 | 54.0 | 79.5 | 1444.3 | 63.0 | 55.6 | 31.4 | 61.6 | 94.0% |
| TRIM (COLING25) | 74.9 | 59.9 | 53.5 | 66.2 | 50.2 | 86.5 | 1443.8 | 63.5 | 51.0 | 32.7 | 61.1 | 92.9% |
| VisionZip (CVPR25) | 76.2 | 58.9 | 56.2 | 67.5 | 58.8 | 82.3 | 1397.1 | 63.3 | 55.6 | 35.8 | 62.4 | 95.7% |
| DART (2025.02) | 75.7 | 59.5 | 56.8 | 67.5 | 57.6 | 81.0 | 1419.5 | 64.2 | 55.7 | 35.7 | 62.5 | 95.8% |
| DivPrune (CVPR25) | 77.2 | 61.1 | 55.6 | 67.7 | 56.2 | 84.7 | 1423.3 | 63.9 | 55.7 | 34.8 | 62.8 | 96.0% |
| **CDPruner (Ours)** | 78.4 | 61.6 | 55.8 | 67.8 | 57.4 | 87.2 | 1453.0 | 65.5 | 55.7 | 37.9 | **64.0** | **98.0%** |
| *Retain 160 Tokens (↓ 94.4%)* | | | | | | | | | | | | |
| PruMerge+ (2024.05) | 70.5 | 56.2 | 57.2 | 66.9 | 50.3 | 71.1 | 1289.6 | 58.0 | 48.9 | 29.3 | 57.3 | 87.7% |
| TRIM (COLING25) | 71.0 | 57.4 | 52.9 | 65.5 | 45.8 | 84.8 | 1275.8 | 61.6 | 45.2 | 29.6 | 57.8 | 87.7% |
| VisionZip (CVPR25) | 71.4 | 55.2 | 55.5 | 67.9 | 55.0 | 74.9 | 1327.8 | 58.6 | 50.4 | 32.3 | 58.8 | 90.0% |
| DART (2025.02) | 72.5 | 56.8 | 56.7 | 67.8 | 54.9 | 75.3 | 1325.4 | 62.0 | 53.6 | 32.2 | 59.8 | 91.7% |
| DivPrune (CVPR25) | 75.0 | 59.3 | 56.1 | 67.1 | 54.1 | 80.0 | 1356.6 | 62.9 | 53.7 | 32.0 | 60.8 | 92.9% |
| **CDPruner (Ours)** | 76.7 | 60.8 | 55.2 | 67.5 | 55.4 | 86.8 | 1425.3 | 64.2 | 53.8 | 36.2 | **62.8** | **96.0%** |

## 4.2 Main results

We first apply CDPruner to LLaVA-1.5, which is widely adopted for evaluating visual token pruning strategies. Table 1 presents the performance of different pruning methods on the LLaVA-1.5-7B model when retaining only 128, 64, or 32 visual tokens. With 77.8% of tokens pruned, CDPruner remarkably maintains nearly all the original performance, surpassing VisionZip by **1.4%**. When the number of visual tokens further decreases to 64, roughly one-tenth of the original token length, attention-based pruning methods exhibit significant performance degradation of **over 25%**, indicating that internal text-visual attention within the language model is not an ideal metric for pruning. Under the same reduction ratio, CDPruner only decreases the original performance by **3.4%**, outperforming VisionZip and DivPrune by **2.6%** and **2.3%**, respectively. With only 5.6% of visual tokens retained, attention and similarity-based methods also encounter noticeable performance degradation because, despite selecting relatively important tokens, they include excessive redundancy and duplication. In this scenario, CDPruner still maintains **94.3%** of the original performance, significantly outperforming the best similarity-based method, DivPrune, by **3%**, which fully demonstrates its effectiveness.

Among all 10 benchmarks, CDPruner achieves particularly strong performance on POPE [Li et al., 2023], even exceeding the unpruned original LLaVA-1.5 model. Since POPE is specifically designed to evaluate visual hallucination, this result suggests that appropriate pruning may help mitigate hallucination in MLLMs, which we believe is a valuable direction for future research. On the other hand, CDPruner shows limited advantage on VizWiz [Gurari et al., 2018], primarily because questions in this benchmark often lack informative context (e.g., "What is this?"), making them insufficiently effective as conditional guidance for the DPP process.

## 4.3 CDPruner for high resolution inputs

Increasing the resolution of input images can improve the performance of MLLMs, but this improvement comes with substantial computational overhead. Higher resolutions introduce more visual tokens, inherently increasing redundancy and thus making it more suitable for pruning. To evaluate this, we apply CDPruner to LLaVA-NeXT, a model specifically designed for handling high-resolution inputs. To ensure a fair comparison by controlling the number of visual tokens, we fix the input resolution to 672×672, resulting in 2,880 visual tokens. As shown in Table 2, with 77.8% of tokens pruned, CDPruner maintains performance comparable to, or slightly better than, the original LLaVA-

Table 3: **Performance comparison of different pruning methods on LLaVA-Video-7B with 64 frames per video.** Here, **Acc.** denotes the average accuracy across 4 video-based benchmarks, and **Rel.** represents the average percentage of performance maintained. Attention-based methods are shown with red background, and similarity-based methods are shown with blue background.

| Method | MLVU | MVBench | LongVideoBench | | | | Video-MME | | | Acc. | Rel. |
| Metric | m-avg | test | val | perception | relation | w/o sub | short | medium | long | | |
|---|---|---|---|---|---|---|---|---|---|---|---|
| *Upper Bound, All 64 × 169 Tokens (100%)* | | | | | | | | | | | |
| LLaVA-Video-7B | 67.7 | 58.2 | 59.0 | 65.0 | 53.8 | 63.6 | 76.6 | 61.2 | 53.1 | 62.1 | 100.0% |
| *Retain 64 × 64 Tokens (↓ 62.1%)* | | | | | | | | | | | |
| FastV(ECCV24) | 63.9 | 55.8 | 56.1 | 60.6 | 52.1 | 61.9 | 73.6 | 59.3 | 52.7 | 59.4 | 95.7% |
| PDrop(CVPR25) | 64.9 | 56.9 | 56.9 | 62.2 | 52.2 | 62.5 | 74.1 | 60.7 | 52.7 | 60.3 | 97.1% |
| SparseVLM(ICML25) | 65.5 | 56.8 | 56.0 | 61.0 | 51.7 | 61.0 | 73.0 | 58.8 | 51.2 | 59.8 | 96.3% |
| DART(2025.02) | 64.1 | 55.5 | 57.5 | 62.1 | 53.5 | 61.6 | 73.0 | 59.9 | 51.9 | 59.7 | 96.1% |
| DivPrune(CVPR25) | 64.1 | 55.1 | 58.6 | 64.2 | 53.7 | 61.1 | 72.9 | 59.3 | 51.2 | 59.7 | 96.2% |
| **CDPruner(Ours)** | **66.3** | **57.4** | **58.7** | 64.5 | 53.7 | **62.6** | 74.6 | 60.3 | 52.9 | **61.3** | **98.6%** |
| *Retain 64 × 32 Tokens (↓ 81.1%)* | | | | | | | | | | | |
| FastV(ECCV24) | 58.5 | 52.7 | 52.4 | 57.0 | 48.5 | 56.0 | 63.8 | 55.9 | 48.4 | 54.9 | 88.5% |
| PDrop(CVPR25) | 59.7 | 53.1 | 52.4 | 56.6 | 48.6 | 58.2 | 67.0 | 57.7 | 50.0 | 55.9 | 89.9% |
| SparseVLM(ICML25) | 60.7 | 54.1 | 53.7 | 58.1 | 49.9 | 59.0 | 69.8 | 56.9 | 50.3 | 56.9 | 91.6% |
| DART(2025.02) | 61.1 | 52.7 | 54.1 | 57.8 | 50.8 | 58.1 | 67.3 | 57.1 | 50.0 | 56.5 | 91.0% |
| DivPrune(CVPR25) | 61.5 | 53.7 | 56.4 | 62.1 | 51.4 | 59.3 | 69.9 | 57.9 | 50.2 | 57.7 | 93.0% |
| **CDPruner(Ours)** | **63.0** | **55.7** | **56.5** | 61.0 | 52.7 | **60.5** | 71.9 | 58.6 | 51.0 | **58.9** | **95.0%** |
| *Retain 64 × 16 Tokens (↓ 90.5%)* | | | | | | | | | | | |
| FastV(ECCV24) | 52.8 | 46.7 | 46.6 | 48.8 | 44.7 | 50.0 | 55.0 | 50.0 | 45.0 | 49.0 | 79.0% |
| PDrop(CVPR25) | 52.8 | 44.3 | 44.3 | 47.5 | 41.4 | 48.9 | 52.9 | 50.0 | 43.8 | 47.6 | 76.5% |
| SparseVLM(ICML25) | 52.0 | 48.7 | 47.6 | 53.0 | 42.8 | 49.8 | 53.8 | 49.3 | 46.3 | 49.5 | 79.9% |
| DART(2025.02) | 56.7 | 50.4 | 51.8 | 56.8 | 47.5 | 55.3 | 64.8 | 52.9 | 48.1 | 53.6 | 86.3% |
| DivPrune(CVPR25) | 58.6 | 52.0 | 52.1 | 57.6 | 47.2 | 56.7 | 67.7 | 54.2 | 48.2 | 54.9 | 88.3% |
| **CDPruner(Ours)** | **58.9** | **53.8** | **52.7** | 57.4 | 48.5 | **57.3** | 66.2 | 56.0 | 49.6 | **55.7** | **89.7%** |

NeXT, demonstrating the higher visual redundancy in high-resolution scenarios. As the reduction ratio further increases to 88.9% and 94.4%, CDPruner still retains up to 98% and 96% of the original performance, outperforming the second-best DivPruner by **2%** and **3.1%**, respectively. These results highlight the strong effectiveness of CDPruner in high-resolution contexts.

## 4.4 CDPruner for video understanding

Video understanding is another task with high visual redundancy. To validate CDPruner in such a scenario, we apply it to LLaVA-Video, an advanced video MLLM. We set the maximum number of video frames to 64, each with a resolution of 384×384, resulting in over 10k tokens and considerable visual redundancy. As demonstrated in Table 3, with 62.1% of visual tokens pruned, CDPruner maintains **98.6%** of the original performance, outperforming PDrop by **1.5%**. As the reduction ratio increases to 81.1%, CDPruner still preserves **95%** performance, significantly exceeding DivPrune's **93%**. Furthermore, when only **16** visual tokens are retained per frame, text-based methods exhibit substantial performance degradation, while CDPruner is able to maintain **89.7%** performance, showing a substantial **10%** improvement over SparseVLM. These results adequately demonstrate the effectiveness of CDPruner in video understanding applications.

## 4.5 CDPruner for advanced architectures

In addition to the LLaVA series, we further apply CDPruner to the most advanced open-source MLLM architectures to validate its generalizability. Here, we select Qwen2.5-VL as a representative model, with the input resolution fixed at 1008×1008, yielding 1,296 visual tokens. Due to the unique structure of its visual encoder and multimodal projector, pruning methods that require the [cls] token are no longer applicable. Therefore, we compare CDPruner only against representative methods from the other two categories, attention-based FastV and similarity-based DivPrune, with results summarized in Table 4. Compared to the LLaVA series, Qwen2.5-VL exhibits a more noticeable performance drop after pruning. This is because visual tokens are already compressed within its projector. Nevertheless, CDPruner consistently outperforms other methods under the same reduction ratios. With 60.5% and 80.2% of tokens pruned, CDPruner retains **97.5%** and **92.8%** of the original performance, surpassing the second-best FastV by **0.5%** and **2.0%**, respectively. When only 128 visual tokens remained, competing methods suffer from severe performance degradation. In contrast, CDPruner maintains **85.2%** of the original performance, significantly higher than DivPrune's **79.9%**, demonstrating the strong generalizability of CDPruner on advanced MLLM architectures.

Table 4: **Performance comparison of different pruning methods on Qwen2.5-VL-7B. Acc.** denotes the average accuracy, **Rel.** represents the average percentage of performance maintained. Attention-based methods are shown with red background, and similarity-based methods with blue.

| Method | TextVQA | ChartQA | AI2D | OCRBench | HallBench | MME | MMB-EN | MMB-CN | Acc. | Rel. |
|---|---|---|---|---|---|---|---|---|---|---|
| *Upper Bound, All 1296 Tokens (100%)* | | | | | | | | | | |
| Qwen2.5-VL-7B | 84.8 | 86.1 | 80.4 | 863 | 46.8 | 2304 | 82.8 | 83.2 | 83.2 | 100.0% |
| *Retain 512 Tokens (↓ 60.5%)* | | | | | | | | | | |
| FastV (ECCV24) | 84.1 | 82.2 | 78.8 | 815 | 42.4 | 2317 | 82.0 | 81.8 | 81.1 | 97.0% |
| DivPrune (CVPR25) | 81.8 | 79.6 | 78.6 | 800 | 43.3 | 2279 | 81.6 | 82.1 | 80.1 | 96.0% |
| **CDPruner** (Ours) | 84.2 | 82.8 | 78.9 | 827 | 42.5 | 2327 | 82.2 | 82.6 | **81.5** | **97.5%** |
| *Retain 256 Tokens (↓ 80.2%)* | | | | | | | | | | |
| FastV (ECCV24) | 81.5 | 70.9 | 76.2 | 703 | 39.0 | 2238 | 79.6 | 78.9 | 76.0 | 90.8% |
| DivPrune (CVPR25) | 76.0 | 65.1 | 76.5 | 692 | 36.4 | 2184 | 80.0 | 79.6 | 74.0 | 88.2% |
| **CDPruner** (Ours) | 82.4 | 73.0 | 77.5 | 749 | 40.1 | 2245 | 80.9 | 79.9 | **77.6** | **92.8%** |
| *Retain 128 Tokens (↓ 90.1%)* | | | | | | | | | | |
| FastV (ECCV24) | 73.8 | 52.2 | 71.4 | 531 | 33.8 | 2008 | 72.9 | 72.2 | 66.2 | 79.0% |
| DivPrune (CVPR25) | 67.0 | 50.4 | 72.1 | 549 | 32.6 | 2108 | 77.8 | 77.8 | 67.3 | 79.9% |
| **CDPruner** (Ours) | 77.8 | 59.2 | 74.0 | 632 | 37.2 | 2127 | 76.2 | 76.5 | **71.3** | **85.2%** |

Table 5: **Performance comparison of different pruning methods on multi-turn dialogues. Acc.** denotes the average accuracy, **Rel.** represents the average percentage of performance maintained. GPT-4o is used for evaluation from six dimensions: Creativity (C), Richness (R), Visual Perception (VP), Logical Coherence (LC), Answer Accuracy (AA), Image Relationship Understanding (IRU).

| Method | C | R | VP | LC | AA | IRU | Acc. | Rel. |
|---|---|---|---|---|---|---|---|---|
| *Upper Bound, All 576 Tokens (100%)* | | | | | | | | |
| LLaVA-1.5-7B | 34.8 | 32.7 | 39.4 | 65.3 | 47.4 | 39.5 | 42.9 | 100.0% |
| *Retain 128 Tokens (↓ 77.8%)* | | | | | | | | |
| TRIM (COLING25) | 35.7 | 34.2 | 38.7 | 64.6 | 46.8 | 39.2 | 42.8 | 99.8% |
| **CDPruner** (Ours) | 36.2 | 34.9 | 40.0 | 66.2 | 48.0 | 40.8 | **44.0** | **102.6%** |
| *Retain 64 Tokens (↓ 88.9%)* | | | | | | | | |
| TRIM (COLING25) | 35.6 | 34.1 | 37.1 | 63.8 | 44.8 | 37.7 | 41.7 | 97.2% |
| **CDPruner** (Ours) | 36.1 | 34.4 | 38.6 | 64.5 | 46.2 | 39.0 | **42.8** | **99.8%** |
| *Retain 32 Tokens (↓ 94.4%)* | | | | | | | | |
| TRIM (COLING25) | 35.4 | 34.0 | 36.1 | 62.8 | 44.2 | 36.9 | 41.2 | 96.0% |
| **CDPruner** (Ours) | 35.6 | 34.0 | 36.7 | 62.9 | 44.6 | 38.0 | **41.5** | **96.7%** |

## 4.6 CDPruner for multi-turn dialog understanding

Multi-turn dialog understanding presents a major challenge for visual token pruning. In single-turn scenarios, pruning methods only need to retain visual tokens most relevant to the current question. In contrast, multi-turn dialogues require preserving more holistic visual semantics to avoid losing information that may be crucial for answering future questions. To evaluate our proposed method, we adopt the MMDU benchmark [Liu et al., 2024f], which includes both multi-turn and multi-image dialogues, and compare our CDPruner against TRIM, which prunes tokens solely based on their relevance to the current query. Both pruning methods are applied to LLaVA-1.5 model for evaluation. To prevent dialogues from exceeding LLaVA's context length limit, we select a subset of 100 samples from MMDU, each containing five dialogue turns and no more than 12 images. GPT-4o is used for evaluation across the six dimensions defined in the original paper.

As shown in Table 5, CDPruner consistently outperforms TRIM across various reduction ratios, demonstrating its superior adaptability to multi-turn dialogues. With 77.8% of visual tokens removed, CDPruner achieves better performance than the LLaVA-1.5 baseline. Even when 88.9% of visual tokens are pruned, it maintains **99.8%** of the original performance, surpassing TRIM by **2.6%**. When only 32 tokens per image are retained, CDPruner still preserves **96.7%** of the original performance. TRIM prunes tokens based on their relevance to the current instruction, which is suboptimal for multi-turn dialogues. If the subsequent question differs significantly from the previous one, the retained tokens may no longer be relevant, resulting in degraded performance. In contrast, CDPruner incorporates diversity modeling via DPP, which enables it to preserve more informative and comprehensive visual content while still considering relevance. As a result, it maintains better performance even in multi-turn scenarios. Exploring a visual token pruning framework that can effectively handle both single- and multi-turn dialogues is a valuable research direction, which we leave for future work.

Table 6: **Efficiency analysis of different pruning methods on LLaVA-NeXT-7B.** The performance is evaluated on POPE. Attention-based methods are shown with red background, attention&similarity-based methods with green background, and similarity-based methods with blue background.

| Method | # Token | FLOPs (T) | Prefill Time (ms/token) | Decode Time (ms/token) | KV Cache (MB) | GPU Memory (GB) | Score (F1) |
|---|---|---|---|---|---|---|---|
| LLaVA-NeXT-7B | 2880 | 41.7 | 246 | 29 | 1440.0 | 16.7 | 86.8 |
| FastV (ECCV24) | 320 | 4.4 (×9.5) | 54 (×4.6) | 23 (×1.2) | 160.3 | 15.6 | 49.5 |
| PDrop (CVPR25) | 320 | 4.5 (×9.3) | 55 (×4.5) | 24 (×1.2) | 160.2 | 15.6 | 60.8 |
| SparseVLM (ICML25) | 320 | 4.5 (×9.3) | 71 (×3.5) | 25 (×1.1) | 161.2 | 18.6 | 76.9 |
| VisionZip (CVPR25) | 320 | **4.2 (×9.9)** | **38 (×6.6)** | **22 (×1.3)** | **160.0** | 14.8 | 82.3 |
| DivPrune (CVPR25) | 320 | **4.2 (×9.9)** | **38 (×6.6)** | **22 (×1.3)** | **160.0** | **13.8** | 84.7 |
| **CDPruner (Ours)** | 320 | **4.2 (×9.9)** | **38 (×6.6)** | **22 (×1.3)** | **160.0** | **13.8** | **87.2** |

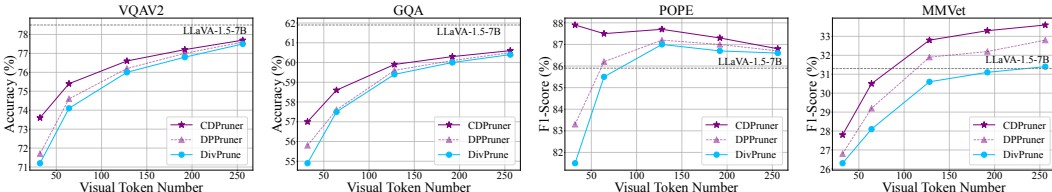

Figure 4: **Ablation study of CDPruner design.** DPPruner denotes applying DPP to visual token pruning without conditioning on instruction relevance, as a degraded variant of CDPruner.

## 4.7 Efficiency analysis

To demonstrate the efficiency of CDPruner, we conduct a comparative analysis against other pruning methods in terms of FLOPs, CUDA latency, KV cache, and GPU memory on the high-resolution MLLM LLaVA-NeXT-7B. All experiments are performed on a single NVIDIA A100-80GB GPU. We choose POPE for evaluating inference efficiency, as it contains questions of similar length and happens to contain only one prefill and one decode stage. As shown in Table 6, when the number of visual tokens is reduced from 2,880 to 320, CDPruner achieves nearly a ×**10** reduction in FLOPs. Regarding CUDA latency, CDPruner reduces the time for prefill and decode stages by ×**6.6** and ×**1.3**, respectively, significantly improving real-world inference efficiency. In addition to runtime latency, CDPruner also reduces KV cache and GPU memory. Compared to all other pruning methods, CDPruner consistently achieves the best efficiency while maintaining the highest performance.

## 4.8 Ablation study

We further conduct an ablation on the design of CDPruner, as illustrated in Figure 4. We compare the performance of different pruning strategies on LLaVA-1.5-7B across four benchmarks, under varying numbers of visual tokens. Here, DPPruner refers to a variant that directly applies DPP to visual token pruning without any condition. This version consistently outperforms DivPrune, demonstrating that the global modeling of token diversity via DPP is more effective than MMDP. When instruction relevance is further incorporated as a condition, CDPruner achieves additional performance gains, validating the benefit of jointly modeling feature similarity and instruction relevance.

## 5 Conclusion

In this paper, we introduce a novel training-free visual token pruning method CDPruner, for MLLM inference acceleration. Specifically, it first defines the conditional similarity between visual tokens based on the instruction, and then reformulates the token pruning problem with DPP to maximize the conditional diversity of the selected subset. Extensive experiments on diverse image and video benchmarks demonstrate that CDPruner achieves state-of-the-art performance across various MLLM architectures, including the LLaVA series and the advanced Qwen2.5-VL. Efficiency analysis further shows that CDPruner significantly reduces inference latency and memory usage while maintaining competitive performance, facilitating the practical deployment of MLLMs in real-world applications.

## Acknowledgements

This work was supported by the National Natural Science Foundation of China (62476011), and by the National Science and Technology Major Project (No. 2022ZD0117800).

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

# Beyond Attention or Similarity: Maximizing Conditional Diversity for Token Pruning in MLLMs

# Appendix

Section A describes the fast greedy MAP inference algorithm used in this work, along with the corresponding pseudocode. Section B provides some details of the experimental setup, including information about model architectures, evaluation benchmarks, comparison methods and implementation. Sections C and D present additional experimental and visualization results respectively. And sections E and F discuss the limitations and broader impacts of this work.

## A Fast greedy MAP inference algorithm for DPP

The direct MAP inference for DPP is NP-hard. Therefore, we adopt the fast greedy algorithm proposed by Chen et al. [2018]. Specifically, given the kernel matrix $\boldsymbol{L} \in \mathbb{R}^{n \times n}$ indexed by elements of $Z$ and the current selected subset $S \subseteq Z$, the next index $j$ to be added in the iteration satisfies:

$$j = \arg\max_{i \in Z \setminus S} \log \det \left( \boldsymbol{L}_{S \cup \{i\}} \right) - \log \det \left( \boldsymbol{L}_S \right). \tag{9}$$

Since $\boldsymbol{L}$ is a PSD matrix, all of its principal minors are also PSD. Suppose $\det(\boldsymbol{L}_S) > 0$, and the Cholesky decomposition $\boldsymbol{L}_S = \boldsymbol{V}\boldsymbol{V}^\top$, where $\boldsymbol{V} \in \mathbb{R}^{|S| \times |S|}$ is an invertible lower triangular matrix. For any $i \in Z \setminus S$, the Cholesky decomposition of $\boldsymbol{L}_{S \cup \{i\}}$ can be derived as:

$$\boldsymbol{L}_{S \cup \{i\}} = \begin{bmatrix} \boldsymbol{L}_S & \boldsymbol{L}_{S,i} \\ \boldsymbol{L}_{i,S} & \boldsymbol{L}_{ii} \end{bmatrix} = \begin{bmatrix} \boldsymbol{V} & \boldsymbol{0} \\ \boldsymbol{c}_i & d_i \end{bmatrix} \begin{bmatrix} \boldsymbol{V} & \boldsymbol{0} \\ \boldsymbol{c}_i & d_i \end{bmatrix}^\top, \tag{10}$$

where the row vector $\boldsymbol{c}_i \in \mathbb{R}^{|S|}$ and scalar $d_i \geq 0$ satisfies:

$$\boldsymbol{V}\boldsymbol{c}_i^\top = \boldsymbol{L}_{S,i}, \tag{11}$$

$$d_i^2 = \boldsymbol{L}_{ii} - \|\boldsymbol{c}_i\|_2^2. \tag{12}$$

In addition, according to eq. (10), it can be derived that

$$\det(\boldsymbol{L}_{S \cup \{i\}}) = \det(\boldsymbol{V}\boldsymbol{V}^\top) \cdot d_i^2 = \det(\boldsymbol{L}_S) \cdot d_i^2. \tag{13}$$

Therefore, eq. (9) is equivalent to

$$j = \arg\max_{i \in Z \setminus S} \log \left( \det(\boldsymbol{L}_S) \cdot d_i^2 \right) - \log \left( \det(\boldsymbol{L}_S) \right) = \arg\max_{i \in Z \setminus S} \log \left( d_i^2 \right). \tag{14}$$

Once eq. (14) is solved, the Cholesky factor of $\boldsymbol{L}_S$ can be efficiently updated after a new item is added to $S$. For each item $i$, $\boldsymbol{c}_i$ and $d_i$ can be updated incrementally. Define $\boldsymbol{c}_i'$ and $d_i'$ as the new vector and scalar of $i \in Z \setminus (S \cup \{j\})$. According to eq. (10) and eq. (11), we have

$$\begin{bmatrix} \boldsymbol{V} & \boldsymbol{0} \\ \boldsymbol{c}_j & d_j \end{bmatrix} \boldsymbol{c}_i'^\top = \boldsymbol{L}_{S \cup \{j\},i} = \begin{bmatrix} \boldsymbol{L}_{S,i} \\ \boldsymbol{L}_{ji} \end{bmatrix}. \tag{15}$$

Combining eq. (15) with eq. (11), we get

$$\boldsymbol{c}_i' = \begin{bmatrix} \boldsymbol{c}_i & (\boldsymbol{L}_{ji} - \langle \boldsymbol{c}_j, \boldsymbol{c}_i \rangle)/d_j \end{bmatrix} \doteq \begin{bmatrix} \boldsymbol{c}_i & e_i \end{bmatrix}. \tag{16}$$

And eq. (12) implies

$$d_i'^2 = \boldsymbol{L}_{ii} - \|\boldsymbol{c}_i'\|_2^2 = \boldsymbol{L}_{ii} - \|\boldsymbol{c}_i\|_2^2 - e_i^2 = d_i^2 - e_i^2. \tag{17}$$

Initially, $S = \emptyset$, and eq. (13) implies $d_i^2 = \det(\boldsymbol{L}_{ii}) = \boldsymbol{L}_{ii}$. The complete algorithm is shown in Algorithm 1. In the $k$-th iteration, for each item $i \in Z \setminus S$, updating $\boldsymbol{c}_i$ and $d_i$ involve the inner product of two vectors of length $k$, resulting in overall complexity $\mathcal{O}(kn)$. Therefore, the greedy algorithm runs in $\mathcal{O}(nm^2)$ time. After parallelizing the for-loop over $i$ using CUDA, the additional inference latency introduced for each sample can be reduced to less than 10ms, which is negligible.

**Algorithm 1** Fast greedy MAP inference algorithm for DPP

---

**Input:** kernel matrix $\boldsymbol{L}$, index list $Z$, retained size $m$
**Output:** selected subset $S$
1: $\boldsymbol{c}_i = [], d_i^2 = \boldsymbol{L}_{ii}$
2: $j = \arg\max_{i \in Z} \log(d_i^2), S = \{j\}$
3: **while** $| S | < m$ **do**
4:     **for** $i \in Z \setminus S$ **do**
5:        $e_i = (\boldsymbol{L}_{ji} - \langle \boldsymbol{c}_j, \boldsymbol{c}_i \rangle)/d_j$
6:        $\boldsymbol{c}_i = [\boldsymbol{c}_i \quad e_i], d_i^2 = d_i^2 - e_i^2$
7:     **end for**
8:     $j = \arg\max_{i \in Z \setminus S} \log(d_i^2), S = S \cup \{j\}$
9: **end while**
10: **return** $S$

---

# B   Details of experimental setup

## B.1   Model architectures

### B.1.1   LLaVA series

**LLaVA-1.5 [Liu et al., 2024a]** The LLaVA series is one of the most widely used open-source vision-language models, and its simple design, low training cost, and outstanding performance have made it a cornerstone in the field of open-source MLLMs. Specifically, the original LLaVA adopts a pretrained CLIP [Radford et al., 2021] as the visual encoder and Vicuna [Chiang et al., 2023] as the language model. A simple linear projector connects these two modules, enabling the LLM to accept image grid features as input. And the visual instruction tuning enables LLaVA to handle vision-language tasks. Compared to the original version, LLaVA-1.5 updates the linear connector with an MLP, increases the input image resolution, and incorporates a broader set of instruction tuning data, resulting in significant performance improvements. The model processes image inputs at a resolution of $336 \times 336$, leading to 576 visual tokens per image.

**LLaVA-NeXT [Liu et al., 2024b]** To further enhance the visual perception capabilities of the model, LLaVA-NeXT, also known as LLaVA-1.6, adopts a dynamic resolution design to increase the input image resolution. Specifically, the model selects the optimal aspect ratio based on the original resolution of the input image, increasing the resolution by up to $4 \times$. Without replacing the visual encoder, LLaVA-NeXT splits the high-resolution image into several sub-images of the same size as the original image. These sub-images are individually encoded and then concatenated as input to the LLM, leading to improved performance in reasoning, OCR, and world knowledge. To ensure a fair comparison by controlling the number of visual tokens, we fix the input resolution to $672 \times 672$, $4 \times$ the original resolution, resulting in totally 2,880 visual tokens.

**LLaVA-Video [Zhang et al., 2024c]** A variant in the LLaVA series specifically designed for video understanding tasks. It introduces the SlowFast representation to balance the number of video frames and the count of visual tokens. The model employs SigLIP [Zhai et al., 2023] as the visual encoder and accepts video inputs with a resolution of $384 \times 384$, encoding each frame into 729 visual tokens. To further reduce computational cost, LLaVA-Video applies $2 \times 2$ average pooling to the grid visual features, reducing the number of visual tokens by $4 \times$. During evaluation, we sample 64 frames per video, resulting in a total of 10,816 visual tokens.

### B.1.2   Advanced architectures

**Qwen2.5-VL [Bai et al., 2025]** The most advanced model of the Qwen-VL series. Building upon its predecessor, Qwen2-VL, it introduces significant enhancements in visual understanding, document parsing, and video comprehension. The model employs a redesigned Vision Transformer architecture with window attention, SwiGLU activation, and RMSNorm, aligning with the Qwen2.5 language model structure. Notably, it supports dynamic resolution and frame rate processing, enabling the comprehension of videos up to an hour long with precise event localization. Qwen2.5-VL excels in tasks such as object detection, OCR, and structured data extraction from documents, making it a versatile visual agent capable of reasoning and tool usage across various domains.

**InternVL3 [Zhu et al., 2025]** One of the most advanced open-source MLLMs at present. Building upon its predecessor, InternVL2.5, it retains the ViT-MLP-LLM architecture, integrating a Vision Transformer with a large language model through an MLP connector. InternVL3 features a native multimodal pre-training paradigm, jointly acquiring linguistic and multimodal capabilities in a single stage. It incorporates Variable Visual Position Encoding to handle extended multimodal contexts and employs advanced training techniques like supervised fine-tuning and mixed preference optimization. InternVL3 demonstrates superior performance across a wide range of multimodal tasks, including tool usage, GUI agents, industrial image analysis, and 3D vision perception.

## B.2 Evaluation benchmarks

### B.2.1 General image benchmarks

**VQAv2 [Goyal et al., 2017]** The second version of the VQA benchmark [Antol et al., 2015] for open-ended visual question answering, designed to evaluate the ability to understand images, natural language, and commonsense knowledge. It includes 265,016 images from COCO [Lin et al., 2014] and abstract scenes, each paired with an average of 5.4 questions. Each question is annotated with 10 ground truth answers and 3 plausible alternatives. We use the test-dev split for evaluation.

**GQA [Hudson and Manning, 2019]** A large-scale visual question answering benchmark built on real images from the Visual Genome dataset [Krishna et al., 2017], designed to assess compositional reasoning and visual understanding. It provides over 22 million balanced question-answer pairs, with each image annotated by a detailed scene graph describing object classes, attributes, and relationships in the scene. We use the test-dev balanced split for evaluation.

**VizWiz [Gurari et al., 2018]** A visual question answering benchmark collected in a real-world accessibility setting, where blind users captured images and asked spoken questions about them. Each visual question is paired with 10 crowdsourced answers. It introduces two key tasks: answering visual questions and predicting whether a question is unanswerable based on the image, highlighting challenges such as poor image quality and ambiguous content. We use the test split for evaluation.

**ScienceQA [Lu et al., 2022]** A large-scale multimodal multiple-choice question answering benchmark focused on diverse scientific domains. It contains 21,208 questions spanning natural science, language science, and social science, categorized into 26 topics, 127 categories, and 379 skills. Among them, 48.7% include image context, 48.2% include text context, and 30.8% include both. A majority of questions are annotated with grounded lectures (83.9%) and detailed explanations (90.5%), offering external knowledge and reasoning to support the correct answer. We use the test split that includes image context (also known as ScienceQA-IMG) for evalution.

**POPE [Li et al., 2023]** A benchmark designed to assess object hallucination in large vision-language models. The images are sourced from COCO [Lin et al., 2014], and the questions focus on whether a specific object is present in the image, assessing the degree of object hallucination. Precision, recall, and F1 score are used to quantify hallucination rates, and we use the test split for evaluation.

**HallusionBench [Guan et al., 2024]** An image-context reasoning benchmark crafted to expose two frequent failure modes of large vision–language models: language hallucination (answers driven by strong linguistic priors that contradict the image) and visual illusion (misleading visual features that produce confident yet wrong responses). Comprising carefully designed examples that remain challenging for GPT-4V and LLaVA-1.5, it enables fine-grained diagnosis of how VLMs over-trust language or under-exploit vision, offering insights for building more faithfully grounded models.

**MME [Fu et al., 2024a]** A comprehensive benchmark evaluating both perception and cognition abilities of multimodal large language models. It contains a total of 14 subtasks. The perception tasks include coarse- and fine-grained recognition as well as OCR. Coarse-grained recognition primarily focuses on the presence, count, position, and color of objects, while fine-grained recognition involves identifying specific posters, celebrities, scenes, landmarks, and artworks. The cognition tasks include commonsense reasoning, numerical calculation, text translation and code reasoning.

**MMBench [Liu et al., 2025]** A comprehensive multimodal benchmark designed to evaluate a wide range of vision-language capabilities. It features a carefully curated dataset with a larger number and greater diversity of evaluation questions and skills compared to existing benchmarks. MMBench also introduces a novel CircularEval strategy, leveraging ChatGPT to convert open-ended model responses into structured choices, enabling more consistent and robust evaluation of model predictions.

**MM-Vet [Yu et al., 2023]** A benchmark focuses on the integration of different multimodal capabilities. It defines 6 core capabilities through 218 challenging examples, including recognition, OCR, knowledge, language generation, spatial awareness, and mathematics. This benchmark utilizes ChatGPT assistant for evaluation, providing unified metrics for assessing answers of varying styles.

### B.2.2 Text-oriented benchmarks

**AI2D [Kembhavi et al., 2016]** A diagram-based question answering benchmark consisting of over 5,000 grade school science diagrams, annotated with more than 150,000 structured labels and ground-truth syntactic parses. It also includes over 15,000 multiple-choice questions aligned with the diagrams, enabling research on visual reasoning and diagram understanding in scientific contexts. We use the test split with mask for evaluation.

**TextVQA [Singh et al., 2019]** A benchmark designed to evaluate a model's ability to read and reason about text within images. The images are primarily sourced from the Open Images v3 dataset Krasin et al. [2017], containing various scenarios such as signs, billboards, and product packaging with rich text information. It introduces a new modality, scene text, that models must recognize and interpret in order to answer questions accurately. This benchmark emphasizes the integration of OCR and visual reasoning for effective multimodal understanding. We use the validation split for evaluation.

**ChartQA [Masry et al., 2022]** A large-scale benchmark designed for question answering over charts, focusing on complex reasoning that involves both visual interpretation and logical or arithmetic operations. It includes 9.6K human-written questions and 23.1K questions generated from chart summaries. Unlike prior template-based benchmarks, ChartQA challenges models to perform multi-step reasoning using both the visual content and underlying data tables of charts, highlighting the need for advanced multimodal understanding. We use the test split for evaluation.

**OCRBench [Liu et al., 2024e]** A comprehensive evaluation benchmark assessing the OCR capabilities of large multimodal models. It comprises 29 datasets across diverse text-related visual tasks, including text recognition, scene text-centric VQA, document-oriented VQA, key information extraction, and handwritten mathematical expression recognition.

### B.2.3 Video benchmarks

**MLVU [Zhou et al., 2024]** The first comprehensive benchmark designed to evaluate multimodal large language models on long video understanding tasks. It features a diverse set of long videos ranging from 3 minutes to 2 hours in length and includes nine evaluation tasks spanning multiple-choice and free-form generation formats. These tasks are categorized into holistic understanding, single-detail understanding, and multi-detail understanding, challenging models to process both global and local information across long video content. We use M-Avg as the evaluation metric.

**MVBench [Li et al., 2024b]** A comprehensive benchmark for evaluating the temporal understanding abilities of multimodal large language models in video comprehension. It consists of 20 challenging tasks specifically designed to require dynamic video analysis beyond single-frame understanding. MVBench introduces a novel static-to-dynamic transformation approach, converting static tasks into temporally grounded ones, thus systematically testing a wide range of temporal reasoning skills from low-level perception to high-level cognition. We use the test split for evaluation.

**LongVideoBench [Wu et al., 2024]** A large-scale benchmark for evaluating long-form video understanding in large multimodal models. It features 3,763 varying-length web-collected videos (up to an hour long) with subtitles, across diverse topics. This benchmark introduces a novel referring reasoning task, where questions include explicit references to specific video segments, requiring models to retrieve and reason over detailed multimodal context. It includes 6,678 human-annotated multiple-choice questions in 17 fine-grained categories. We use the validation split for evaluation.

**Video-MME [Fu et al., 2024b]** The first comprehensive benchmark specifically designed to evaluate multimodal large language models in video understanding. It includes 900 manually selected and annotated videos totaling 256 hours, covering 6 primary domains and 30 subfields. This benchmark supports diverse temporal lengths (from 11 seconds to 1 hour) and integrates multiple modalities such as video frames, subtitles, and audio. With 2,700 expert-annotated question-answer pairs, Video-MME provides a high-quality, fine-grained assessment of MLLMs' ability to reason over complex sequential and multimodal information. We do not use subtitles during evaluation.

### B.3 Comparison methods

#### B.3.1 Text-based methods

**FastV [Chen et al., 2024a]** The first work to identify the inefficient visual attention phenomena in MLLMs. Based on this observation, FastV proposes a straightforward solution, that is, to prune the part of visual tokens with the lowest visual-text attention score after layer 2 of the model, thereby achieving MLLM inference acceleration in a training-free manner.

**PyramidDrop [Xing et al., 2024]** Building upon FastV, PyramidDrop further observes that pruning in shallow layers has a larger impact on model performance, while the redundancy of visual tokens tends to increase with model depth. Based on this insight, it proposes a hierarchical pruning strategy that divides the MLLM into multiple stages and prunes a certain proportion of visual tokens at the end of each stage, leading to improved performance.

**SparseVLM [Zhang et al., 2024b]** Similar to PyramidDrop, SparseVLM also adopts a multi-stage token pruning strategy. However, unlike previous approaches, it focuses on the impact of the instruction tokens on vision-language attention. It argues that not all text tokens contribute to the visual token pruning, only those highly relevant to the visual content are important. Therefore, it first selects the text tokens most related to the visual input as raters, and uses their attention to the visual tokens to guide the pruning process, leading to further performance improvements.

#### B.3.2 Vision-based methods

**LLaVA-Prumerge [Shang et al., 2024]** The first work to perform token pruning solely based on visual information. It first selects important visual tokens using attention scores from the visual encoder, and then merges each of the remaining tokens with its most similar selected token through a clustering-based approach. Building on this, LLaVA-Prumerge+ introduces spatially uniform sampling to further enhance performance.

**TRIM [Song et al., 2024]** Pruning only based on visual information while ignoring user instructions may lead to suboptimal performance. TRIM addresses this by leveraging CLIP metrics for pruning. Specifically, it computes the cosine similarity between image tokens from the visual encoder and text tokens from the text encoder, and uses these similarities to estimate the importance of each visual token. Tokens with lower similarity scores are pruned to accelerate inference.

**VisionZip [Yang et al., 2024b]** Similar to LLaVA-Prumerge, VisionZip also relies on visual information for token pruning. It observes that attention within the visual encoder is highly concentrated, and therefore first selects several dominant tokens based on visual attention. Then, among all the remaining tokens, a set of contextual tokens is obtained through clustering. These two groups are combined and fed into the language model, aiming to preserve as much visual information as possible.

#### B.3.3 Similarity-based methods

**DART [Wen et al., 2025b]** This work argues that in token pruning, duplication matters more than importance. Based on this insight, it first selects a small set of pivot tokens, and then iteratively retains the most diverse tokens from the remaining ones by selecting those with the lowest similarity to the already selected tokens. Finally, a group of the most diverse visual tokens is obtained.

**DivPrune [Alvar et al., 2025]** This work also focuses on token diversity. However, unlike previous approaches, DivPrune reformulates the token pruning problem as a MMDP, aiming to retain the most diverse subset by maximizing the minimum pairwise distance among the selected tokens.

### B.4 Implementation details

For image benchmarks, we use the official implementation of LLaVA[3]. For video benchmarks, we adopt the official codebase of LLaVA-NeXT[4] for the model architecture and utilize lmms-eval[5] for evaluation. For advanced architectures like Qwen2.5-VL, we employ VLMEvalKit[6] for evaluation.

---

[3]https://github.com/haotian-liu/LLaVA
[4]https://github.com/LLaVA-VL/LLaVA-NeXT
[5]https://github.com/EvolvingLMMs-Lab/lmms-eval
[6]https://github.com/open-compass/VLMEvalKit

Table 7: **Performance comparison of different pruning methods on LLaVA-1.5-13B. Acc.** denotes the average performance across 10 benchmarks, **Rel.** represents the average percentage of performance maintained. Attention-based methods are shown with red background, attention&similarity-based methods with green background, and similarity-based methods with blue background.

| Method | VQA$^{V2}$ | GQA | VizWiz | SQA$^{IMG}$ | VQA$^{Text}$ | POPE | MME | MMB$^{EN}$ | MMB$^{CN}$ | MMVet | Acc. | Rel. |
|---|---|---|---|---|---|---|---|---|---|---|---|---|
| *Upper Bound, All 576 Tokens (100%)* | | | | | | | | | | | | |
| LLaVA-1.5-13B | 80.0 | 63.3 | 53.6 | 72.8 | 61.2 | 86.0 | 1531.2 | 68.5 | 63.5 | 36.2 | 66.2 | 100.0% |
| *Retain 128 Tokens (↓ 77.8%)* | | | | | | | | | | | | |
| FastV(ECCV24) | 75.3 | 58.3 | 54.6 | 74.2 | 58.6 | 75.5 | 1460.6 | 66.1 | 62.3 | 32.8 | 63.1 | 95.4% |
| PDrop(CVPR25) | 78.2 | 61.0 | 53.8 | 73.3 | 60.2 | 83.6 | 1489.5 | 67.5 | 62.8 | 32.1 | 64.7 | 97.4% |
| SparseVLM(ICML25) | 77.6 | 59.6 | 51.4 | 74.3 | 59.3 | 85.0 | 1487.9 | 68.4 | 62.6 | 35.2 | 64.8 | 97.8% |
| PruMerge+(2024.05) | 76.2 | 58.3 | 52.8 | 73.3 | 56.1 | 82.7 | 1445.9 | 66.3 | 61.2 | 33.6 | 63.3 | 95.5% |
| TRIM(COLING25) | 76.4 | 59.4 | 49.7 | 72.4 | 55.0 | 86.8 | 1426.9 | 67.1 | 58.4 | 35.1 | 63.2 | 95.2% |
| VisionZip(CVPR25) | 76.8 | 57.9 | 52.3 | 73.8 | 58.9 | 82.7 | 1449.2 | 67.4 | 62.5 | 36.0 | 64.1 | 97.0% |
| DART(2025.02) | 75.7 | 57.7 | 53.0 | 74.2 | 58.7 | 80.4 | 1395.0 | 65.4 | 62.2 | 34.8 | 63.2 | 95.7% |
| DivPrune(CVPR25) | 77.1 | 59.2 | 53.5 | 72.8 | 58.0 | 86.8 | 1457.7 | 66.3 | 60.7 | 34.4 | 64.2 | 96.8% |
| **CDPruner(Ours)** | 77.7 | 59.7 | 52.9 | 73.2 | 58.4 | 87.3 | 1478.0 | 67.5 | 61.5 | 36.2 | **64.8** | **98.0%** |
| *Retain 64 Tokens (↓ 88.9%)* | | | | | | | | | | | | |
| FastV(ECCV24) | 65.3 | 51.9 | 53.8 | 73.1 | 53.4 | 56.9 | 1246.4 | 59.2 | 55.1 | 26.9 | 55.8 | 84.7% |
| PDrop(CVPR25) | 70.8 | 54.1 | 50.5 | 73.1 | 55.3 | 66.1 | 1247.0 | 63.1 | 56.6 | 21.9 | 57.4 | 85.9% |
| SparseVLM(ICML25) | 73.2 | 55.9 | 52.1 | 73.0 | 57.1 | 77.9 | 1374.3 | 65.2 | 60.3 | 32.9 | 61.6 | 93.2% |
| PruMerge+(2024.05) | 72.6 | 56.3 | 52.4 | 73.5 | 54.4 | 75.7 | 1338.2 | 65.0 | 59.3 | 30.3 | 60.6 | 91.5% |
| TRIM(COLING25) | 73.2 | 57.9 | 49.2 | 72.0 | 52.0 | 86.5 | 1406.2 | 65.0 | 52.7 | 27.8 | 60.7 | 90.6% |
| VisionZip(CVPR25) | 73.7 | 56.2 | 53.2 | 74.2 | 57.4 | 75.7 | 1379.6 | 64.9 | 61.3 | 33.4 | 61.9 | 93.8% |
| DART(2025.02) | 72.4 | 55.7 | 53.4 | 73.8 | 57.3 | 72.8 | 1380.0 | 64.7 | 60.6 | 32.8 | 61.3 | 92.8% |
| DivPrune(CVPR25) | 75.2 | 57.9 | 54.4 | 71.7 | 57.4 | 84.5 | 1454.2 | 64.1 | 59.8 | 29.3 | 62.7 | 94.1% |
| **CDPruner(Ours)** | 76.7 | 59.4 | 53.6 | 72.5 | 57.6 | 87.1 | 1466.8 | 65.5 | 58.8 | 35.2 | **64.0** | **96.6%** |
| *Retain 32 Tokens (↓ 94.4%)* | | | | | | | | | | | | |
| PruMerge+(2024.05) | 66.8 | 54.1 | 52.3 | 71.7 | 52.4 | 67.4 | 1269.1 | 61.1 | 53.5 | 28.7 | 57.1 | 86.5% |
| TRIM(COLING25) | 69.8 | 55.6 | 48.8 | 70.4 | 49.6 | 85.8 | 1284.7 | 61.1 | 45.4 | 26.4 | 57.9 | 86.4% |
| VisionZip(CVPR25) | 68.4 | 52.7 | 53.0 | 72.9 | 55.2 | 66.8 | 1257.7 | 61.2 | 55.8 | 29.3 | 57.8 | 87.6% |
| DART(2025.02) | 68.1 | 53.9 | 52.0 | 73.2 | 55.1 | 66.9 | 1282.8 | 61.9 | 56.2 | 29.4 | 58.1 | 88.0% |
| DivPrune(CVPR25) | 72.0 | 56.2 | 54.5 | 70.9 | 54.6 | 79.3 | 1405.2 | 61.7 | 57.2 | 27.8 | 60.4 | 90.8% |
| **CDPruner(Ours)** | 75.2 | 58.5 | 53.5 | 71.9 | 55.3 | 87.6 | 1421.0 | 63.7 | 56.6 | 30.9 | **62.4** | **93.8%** |

# C  Additional experimental results

## C.1  CDPruner for larger language model

To evaluate the effectiveness of our proposed method on larger language models, we apply CDPruner to two models equipped with 13B LLMs: LLaVA-1.5-13B and LLaVA-NeXT-13B. The results are presented in Tables 7 and 8. The larger language models lead to significant performance improvements and also make MLLMs less sensitive to visual token pruning. Among various pruning strategies, text-attention based methods benefit the most from scaling up the language model, indicating that larger LLM brings more accurate attention. Across different types of pruning methods, CDPruner consistently outperforms all other approaches under various reduction ratios. With 77.8% of visual tokens removed, our method retains 98.0% and 99.9% of the original performance on LLaVA-1.5-13B and LLaVA-NeXT-13B, respectively, demonstrating its effectiveness on larger language models.

## C.2  CDPruner for advanced open-source MLLM

In addition to Qwen2.5-VL, we further apply CDPruner to one of the most advanced open-source MLLMs to date, InternVL3. The results are shown in Table 9. Here, we fix the input resolution to 896×896, yielding 1,280 visual tokens. Notably, unlike its performance on the LLaVA series, DivPrune exhibits a significant performance drop on InternVL3, as it does not account for the relevance to user instructions during pruning. In contrast, our CDPruner jointly considers both diversity and relevance, consistently achieving the best performance across different reduction ratios. Specifically, even when 90% of the visual tokens are removed, our method retains 83.9% of the original performance, 3% higher than the second-best FastV, demonstrating its effectiveness and adaptability in advanced MLLM architectures.

## C.3  Efficiency analysis on larger language model

Here, we conduct an additional efficiency analysis on LLaVA-NeXT-13B, which has higher computational demands. As shown in Table 10, the combination of higher input image resolution and a larger LLM results in significantly increased inference cost. Our CDPruner effectively reduces the number of visual tokens from 2,880 to 320, achieving a 10× reduction in FLOPs, along with 6.8× and 1.4×

Table 8: **Performance comparison of different pruning methods on LLaVA-NeXT-13B. Acc.** denotes the average performance across 10 benchmarks, **Rel.** represents the average percentage of performance maintained. Attention-based methods are shown with red background, attention&similarity-based methods with green background, and similarity-based methods with blue background.

| Method | VQA$^{V2}$ | GQA | VizWiz | SQA$^{IMG}$ | VQA$^{Text}$ | POPE | MME | MMB$^{EN}$ | MMB$^{CN}$ | MMVet | Acc. | Rel. |
|---|---|---|---|---|---|---|---|---|---|---|---|---|
| | *Upper Bound, All 2880 Tokens (100%)* | | | | | | | | | | | |
| LLaVA-NeXT-7B | 82.3 | 64.4 | 59.1 | 73.1 | 63.2 | 85.3 | 1539.5 | 68.5 | 61.2 | 45.0 | 67.9 | 100.0% |
| | *Retain 640 Tokens (↓ 77.8%)* | | | | | | | | | | | |
| FastV (ECCV24) | 79.4 | 60.9 | 56.4 | 71.7 | 60.7 | 80.2 | 1516.7 | 65.5 | 59.9 | 43.8 | 65.4 | 96.4% |
| PDrop (CVPR25) | 81.1 | 62.8 | 58.1 | 71.7 | 62.1 | 84.4 | 1559.1 | 66.6 | 60.8 | 39.7 | 66.5 | 97.6% |
| SparseVLM (ICML25) | 79.9 | 62.7 | 57.5 | 72.5 | 62.8 | 85.6 | 1562.7 | 68.8 | 64.0 | 41.3 | 67.3 | 98.9% |
| PruMerge+ (2024.05) | 78.7 | 62.8 | 56.2 | 70.6 | 56.2 | 83.7 | 1497.3 | 67.4 | 61.9 | 39.4 | 65.2 | 95.6% |
| TRIM (COLING25) | 79.4 | 63.1 | 54.1 | 71.2 | 57.6 | 87.3 | 1554.6 | 68.7 | 61.2 | 42.3 | 66.3 | 97.2% |
| VisionZip (CVPR25) | 79.7 | 62.9 | 56.2 | 70.8 | 62.1 | 85.8 | 1549.2 | 68.1 | 62.6 | 46.8 | 67.2 | 99.2% |
| DART (2025.02) | 79.3 | 62.7 | 56.2 | 71.0 | 61.3 | 85.2 | 1542.4 | 67.6 | 61.9 | 45.5 | 66.8 | 98.4% |
| DivPrune (CVPR25) | 80.4 | 63.5 | 56.7 | 72.2 | 59.2 | 86.5 | 1526.1 | 67.5 | 62.9 | 39.0 | 66.4 | 97.3% |
| **CDPruner (Ours)** | 81.0 | 64.0 | 57.1 | 71.8 | 61.0 | 87.5 | 1545.6 | 68.9 | 62.1 | 47.3 | **67.8** | **99.9%** |
| | *Retain 320 Tokens (↓ 88.9%)* | | | | | | | | | | | |
| FastV (ECCV24) | 669.8 | 54.6 | 53.3 | 70.5 | 55.4 | 63.6 | 1279.0 | 59.8 | 54.4 | 30.2 | 57.6 | 84.5% |
| PDrop (CVPR25) | 75.4 | 57.7 | 52.1 | 72.1 | 56.2 | 74.6 | 1386.3 | 62.8 | 55.3 | 29.5 | 60.5 | 88.2% |
| SparseVLM (ICML25) | 76.7 | 60.9 | 54.7 | 70.9 | 60.0 | 81.5 | 1491.6 | 68.0 | 63.5 | 39.3 | 65.0 | 95.5% |
| PruMerge+ (2024.05) | 75.9 | 61.1 | 53.6 | 70.7 | 55.9 | 79.1 | 1426.5 | 66.6 | 60.6 | 36.5 | 63.1 | 92.6% |
| TRIM (COLING25) | 75.9 | 61.3 | 52.2 | 69.9 | 52.8 | 87.2 | 1476.6 | 67.3 | 57.4 | 33.1 | 63.1 | 91.9% |
| VisionZip (CVPR25) | 76.8 | 60.7 | 54.8 | 70.2 | 60.7 | 82.3 | 1487.3 | 66.5 | 62.3 | 41.1 | 65.0 | 95.6% |
| DART (2025.02) | 76.4 | 60.9 | 54.2 | 69.8 | 59.7 | 81.1 | 1457.4 | 65.9 | 61.9 | 41.4 | 64.4 | 94.8% |
| DivPrune (CVPR25) | 78.1 | 61.8 | 55.0 | 72.3 | 57.6 | 85.2 | 1473.0 | 65.9 | 61.9 | 39.2 | 65.1 | 95.4% |
| **CDPruner (Ours)** | 79.6 | 63.1 | 55.1 | 71.6 | 58.7 | 87.6 | 1498.5 | 66.3 | 61.8 | 42.4 | **66.1** | **97.1%** |
| | *Retain 160 Tokens (↓ 94.4%)* | | | | | | | | | | | |
| PruMerge+ (2024.05) | 71.6 | 57.9 | 50.8 | 70.1 | 52.8 | 72.1 | 1345.9 | 63.2 | 57.1 | 30.6 | 59.3 | 86.8% |
| TRIM (COLING25) | 72.1 | 58.9 | 51.2 | 69.1 | 49.2 | 87.0 | 1392.3 | 64.7 | 51.6 | 27.8 | 60.2 | 87.3% |
| VisionZip (CVPR25) | 72.4 | 57.8 | 52.5 | 69.7 | 58.6 | 76.8 | 1393.9 | 64.8 | 60.0 | 35.9 | 61.8 | 90.8% |
| DART (2025.02) | 72.8 | 58.7 | 52.1 | 70.1 | 57.2 | 75.7 | 1389.3 | 64.6 | 60.8 | 35.0 | 61.6 | 90.5% |
| DivPrune (CVPR25) | 75.6 | 60.0 | 53.5 | 71.4 | 56.3 | 81.9 | 1436.7 | 65.1 | 60.9 | 37.4 | 63.4 | 92.9% |
| **CDPruner (Ours)** | 77.8 | 62.2 | 53.1 | 71.7 | 56.7 | 88.3 | 1476.9 | 65.9 | 60.1 | 40.4 | **65.0** | **95.2%** |

Table 9: **Performance comparison of different pruning methods on InternVL3-8B. Acc.** denotes the average accuracy, **Rel.** represents the average percentage of performance maintained. Attention-based methods are shown with red background, and similarity-based methods with blue.

| Method | AI2D | TextVQA | ChartQA | OCRBench | HallBench | MME | MMB-EN | MMB-CN | Acc. | Rel. |
|---|---|---|---|---|---|---|---|---|---|---|
| | *Upper Bound, All 1280 Tokens (100%)* | | | | | | | | | |
| InternVL3-8B | 85.2 | 81.5 | 85.1 | 853 | 50.0 | 2394 | 83.9 | 82.6 | 84.2 | 100.0% |
| | *Retain 256 Tokens (↓ 80.0%)* | | | | | | | | | |
| FastV (ECCV24) | 82.2 | 74.4 | 70.7 | 632 | 48.5 | 2348 | 83.6 | 82.0 | 77.8 | 92.4% |
| DivPrune (CVPR25) | 80.9 | 64.7 | 57.5 | 477 | 38.7 | 2249 | 80.8 | 80.2 | 70.4 | 82.8% |
| **CDPruner (Ours)** | 82.7 | 75.7 | 72.0 | 640 | 48.8 | 2334 | 83.5 | 81.7 | **78.1** | **92.9%** |
| | *Retain 128 Tokens (↓ 90.0%)* | | | | | | | | | |
| FastV (ECCV24) | 77.3 | 63.7 | 46.9 | 426 | 42.5 | 2250 | 81.3 | 80.2 | 68.4 | 80.9% |
| DivPrune (CVPR25) | 76.4 | 55.6 | 42.7 | 378 | 37.7 | 2166 | 78.4 | 77.6 | 64.3 | 75.7% |
| **CDPruner (Ours)** | 79.9 | 67.5 | 50.8 | 471 | 44.6 | 2282 | 82.1 | 80.3 | **70.8** | **83.9%** |

decreases in prefill and decode time, respectively. Meanwhile, it maintains competitive performance, demonstrating the efficiency of CDPruner for larger MLLM inference.

### C.4 Ablation study on balance factor

Since the amount of information contained in the instructions of different benchmarks varies, we can introduce a balance factor $\theta$ to control the trade-off between diversity and relevance. Specifically, from eq. (8), we derive $\alpha = \theta/(2(1 - \theta))$, which is then used to transform the relevance vector $\tilde{r}$ and construct a new conditional kernel matrix:

$$\tilde{L}' = \text{diag}\left(\exp\left(\alpha\tilde{r}\right)\right) \cdot L \cdot \text{diag}\left(\exp\left(\alpha\tilde{r}\right)\right). \tag{18}$$

The updated log-probability of a subset S for DPP is given by:

$$\log\det\left(\tilde{L}_S\right) = 2\alpha \cdot \sum_{i\in S} \tilde{r}_i + \log\det\left(L_S\right) \propto \theta \cdot \sum_{i\in S} \tilde{r} + (1 - \theta) \cdot \log\det\left(L_S\right) \tag{19}$$

By adjusting $\theta$, we can modulate the relative importance of relevance and diversity in the modeling process. As shown in Table 11, the ablation results for $\theta$ indicate that the optimal value varies across benchmarks. Selecting the best factor value for each dataset leads to performance improvements. It is worth noting that even without introducing this balancing factor (i.e., the version used in the main

Table 10: **Efficiency analysis of different pruning methods on LLaVA-NeXT-13B.** The performance is evaluated on POPE. Attention-based methods are shown with red background, attention&similarity-based methods with green background, and similarity-based methods with blue background.

| Method | # Token | FLOPs (T) | Prefill Time (ms/token) | Decode Time (ms/token) | KV Cache (MB) | GPU Memory (GB) | Score (F1) |
|---|---|---|---|---|---|---|---|
| LLaVA-NeXT-13B | 2880 | 79.9 | 434 | 44 | 2250.0 | 30.1 | 85.3 |
| FastV (ECCV24) | 320 | 8.5 (×9.4) | 75 (×5.8) | 33 (×1.3) | 250.8 | 28.0 | 63.6 |
| PDrop (CVPR25) | 320 | 8.5 (×9.4) | 86 (×5.0) | 34 (×1.3) | 250.7 | 28.0 | 74.6 |
| SparseVLM (ICML25) | 320 | 8.5 (×9.4) | 101 (×4.3) | 38 (×1.2) | 254.8 | 31.6 | 81.5 |
| VisionZip (CVPR25) | 320 | **8.2 (×9.7)** | **64 (×6.8)** | **32 (×1.4)** | **250.0** | 26.6 | 82.3 |
| DivPrune (CVPR25) | 320 | **8.2 (×9.7)** | **64 (×6.8)** | **32 (×1.4)** | **250.0** | **25.9** | 85.2 |
| **CDPruner (Ours)** | 320 | **8.2 (×9.7)** | **64 (×6.8)** | **32 (×1.4)** | **250.0** | **25.9** | **87.6** |

Table 11: **Ablation study of balance factor on LLaVA-1.5-7B, 64 visual tokens retained.**

| $\theta$ | VQA$^{\text{V2}}$ | GQA | VizWiz | SQA$^{\text{IMG}}$ | VQA$^{\text{Text}}$ | POPE | MME | MMB$^{\text{EN}}$ | MMB$^{\text{CN}}$ | MMVet | Acc. | Rel. |
|---|---|---|---|---|---|---|---|---|---|---|---|---|
| 0.0 | 74.6 | 57.6 | 53.9 | 67.9 | 55.8 | 86.2 | 1358.7 | 59.3 | 53.4 | 29.2 | 60.6 | 95.7% |
| 0.2 | 74.8 | 58.2 | 53.8 | 68.2 | 55.7 | 86.6 | 1362.1 | 59.4 | 53.3 | 29.3 | 60.7 | 95.9% |
| 0.4 | 75.1 | 58.7 | 53.9 | 68.1 | 55.7 | 87.2 | 1378.2 | 59.5 | 53.0 | 29.5 | 61.0 | 96.2% |
| 0.6 | 75.5 | 58.9 | 54.1 | 68.5 | 55.6 | 87.3 | 1396.3 | 60.3 | 52.9 | 30.7 | 61.4 | 97.0% |
| 0.8 | 75.2 | 58.5 | 53.3 | 68.4 | 55.0 | 87.4 | 1415.3 | 61.6 | 52.8 | 29.4 | 61.2 | 96.5% |
| **best** | 75.5 | 58.9 | 54.1 | 68.5 | 55.8 | 87.4 | 1415.3 | 61.6 | 53.4 | 30.7 | **61.7** | **97.5%** |

experiments), our method already achieves strong results. Therefore, in practical applications, one may choose whether to introduce and tune this additional hyperparameter based on specific needs.

## D  Additional visualization results

Here, we provide additional visualizations of relevance scores in Figure 5. It can be clearly observed that models with language-image pre-training are able to effectively capture the correspondence between user instructions and regions of interest in the image, which is crucial for instruction-guided visual token pruning in multimodal large language models.

## E  Limitations

One limitation of our work is that the proposed method can only be applied to open-source MLLMs, where the encoded visual tokens can be accessed during inference. However, there exist many black-box models, including ChatGPT, Gemini, and Claude, which also require significant computational resources for visual reasoning. Moreover, although our method is applicable to state-of-the-art open-source MLLM architectures such as Qwen2.5-VL and InternVL3, and achieves superior performance compared to existing approaches, these models are generally more sensitive to visual token pruning. It can be observed that, compared to the LLaVA series, these advanced models tend to suffer greater performance degradation after pruning. This is likely due to the fact that their architectures already incorporate visual token compression techniques like pixel unshuffle. Exploring how to enable efficient inference within these architectures will be an important direction of our future work.

## F  Broader impacts

Recently, MLLMs have been widely applied across various industries, thanks to their powerful reasoning capabilities. However, redundant visual inputs bring high computational complexity and significantly increases its usage cost. In this work, we propose a simple yet effective solution that accelerates MLLM inference by visual token pruning without the need of any additional training. We believe this approach can facilitate the practical application of MLLMs by reducing deployment costs, lowering inference latency, and enabling usage on resource-constrained edge devices. It is important to note that this work does not mitigate the potential misuse of MLLMs by malicious actors.

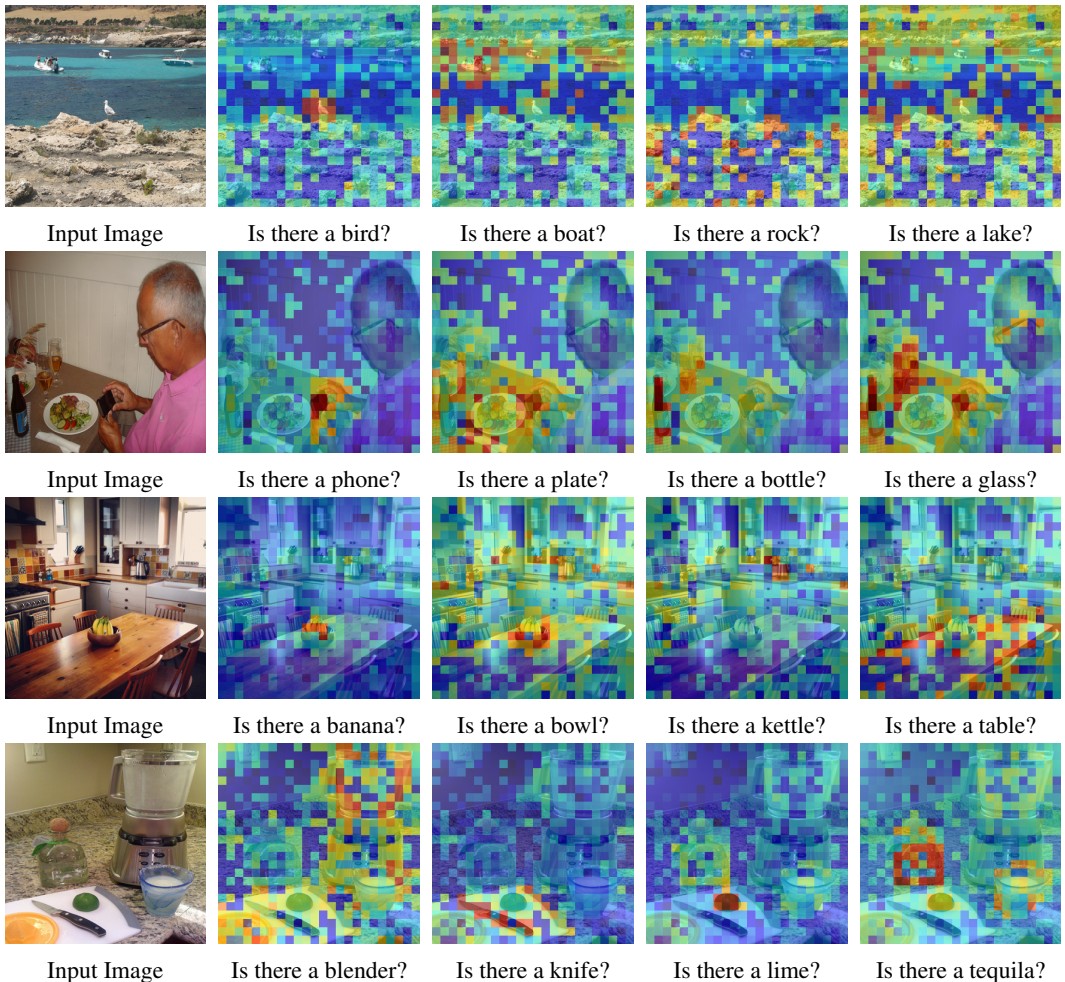

Figure 5: **Additional visualizations of relevance scores.** We compute the relevance scores for several samples from the POPE benchmark using LLaVA-1.5-7B, with the instruction following the template: "Is there a {object} in the image?" **Red** indicates high relevance, while **blue** indicates low.

