# OpenReview forum: "Beyond Attention or Similarity: Maximizing Conditional Diversity for Token Pruning in MLLMs"
_NeurIPS.cc/2025/Conference — NeurIPS 2025 poster_

### Official Review · Reviewer_4nPZ · 2025-06-22

**Clarity:** 3
**Significance:** 4
**Originality:** 3
**Rating:** 5
**Confidence:** 3

**Summary:**

This paper proposes CDPruner, a model-agnostic approach to condense sequence length in multimodal LLMs  by maximizing diversity of the retained tokens conditional on the query and the other retained tokens.

**Questions:**

- Runtime: Can you expand on runtime at the beginning of the introduction where you mention the relate improvements in FLOPs and memory? Or even better add an extra column in one of the result tables comparing runtime of all methods. This would make this work even more comprehensive
- Clarify better how you arrive at K tokens from a given input size. I thought this is a controllable hyperparameter but in multiple occasions  it sounds like it is static ("we fix the input resolution to 672×672, resulting in 2,880 visual tokens" and similarly later for videos)
Abbreviating Large Language Models as LMMs in the first sentence gave me a good laugh

**Ethical Concerns:**

["NO or VERY MINOR ethics concerns only"]

**Final Justification:**

I am convinced that this is a solid and impactful paper and thus retained my original score of 5 through the rebuttal phase.

**Limitations:**

Yes

**Quality:**

4

**Strengths And Weaknesses:**

Positive
- Pruning is conditional on the question, unlike for most (all?) attention or similarity based pruning methods
- CDPruner is rigorously  compared against half a dozen of other methods on 14 datasets for both visual Q&A and video understanding
- Performance is consistently superior in all settings
- Method is novel

Negatives:
- The method is presented generically for MLLMs but it seems that it is limited to Image/Video QA
- The "Rel." metric seems wrong or unintuitive. Table 2, it seems that it is normalized to the first row, but e.g., 2nd row 62.2/65.2 = 0.954 (not 0.956) and especially the CDPruner value of 100.1% does not make sense

---

> ### Author Rebuttal · Authors · 2025-07-30
>
> We sincerely thank the reviewer for the thorough review and valuable time. We are encouraged by the recognition of our method as novel and our performance as superior. After carefully considering the comments, we provide the following responses:
> 1. **Evaluation benchmarks**:
>
>    We follow the most commonly adopted benchmarks used in open-source MLLMs, covering **general visual question answering, document understanding and OCR, as well as video understanding**. We plan to extend our evaluation to a broader range of tasks in future work.
>
> 2. **Calculation of the “Rel.” metric**:
>
>    The “Rel.” metric is calculated as the average of the relative accuracies for each benchmark, rather than using a single relative average accuracy across all tasks. Thanks for pointing this out — we will clarify the calculation method of the metrics in the revised version.
>
> 3. **Runtime**:
>
>    We have already described the runtime in **L18** of the abstract and **L57** of the introduction. Here, the reported **CUDA latency** refers to the actual runtime during inference. Furthermore, we provide a detailed analysis of runtime and efficiency in **Section 4.6** and **Table 5**. Following the reviewer’s suggestion, we will include runtime latency as an additional column in Tables 1–4 in the revised version.
>
> 4. **Explanation of token number $K$**:
>
>    The number of retained tokens $K$ here is not a hyperparameter but rather an input to the algorithm. The goal of our work is to improve the inference efficiency of MLLMs to meet various deployment constraints. Therefore, $K$ can be set according to the available computational budget. For example, the original LLaVA-NeXT-7B model has an inference latency of approximately 280 ms per sample. In latency-sensitive scenarios requiring < 60 ms inference time, the number of visual tokens must be reduced from 2880 to 320. In this case, we should set $K \le 320$. In short, users can flexibly set the retained token number $K$ based on their specific computational constraints, and our CDPruner is capable of adapting to different values of $K$ while consistently achieving strong performance. The values of $K$ presented in the tables are intended to illustrate performance under different reduction ratios, following previous works. The high-resolution input of 2880 visual tokens, as well as the token number in video scenarios, are fixed for fair comparison and do not directly determine the value of $K$.
>
> 5. **Abbreviation typo**:
>
>    We thank the reviewer for the careful review. We will correct this typo in the revised version.

---

> > ### Comment · Reviewer_4nPZ · 2025-08-01
> > **Performance metrics**
> >
> > Thanks for the responses, it's more clear now. CUDA latency is only a subset of entire algorithm runtime, hence I asked to have this additional information.
> >
> > Regarding the Rel. metric: Table 1/2 still seem wrong: Accuracy is stated to be an average across 10 datasets but MME is not given in accuracy. So: Which metric is used for MME? The average of the accuracies of the 9 other datasets is not what is listed as "Accuracy"? This lets me to believe that MME may give you accuracy as a secondary metric which you do not report in the Table but use to calculate the average accuracy?

---

> > > ### Author Response · Authors · 2025-08-01
> > > **Reply to performance metrics**
> > >
> > > We thank the reviewer for the comments. Regarding the efficiency analysis, in addition to the overall inference latency, we have conducted a detailed analysis of the running time for the DPP part in our response to Reviewer w7zd, and we have also explained the additional cost introduced by text feature extraction in our response to Reviewer 5bbk's Q1. We respectfully invite the reviewer to refer to those responses for more details.
> > >
> > > As for the MME benchmark in the “Rel.” metric, according to the official paper, the entire MME benchmark consists of two parts: perception task, which includes 10 subtasks totaling 2000 points, and cognition task, which includes 4 subtasks totaling 800 points. In Tables 1/2, we follow the setting of the original LLaVA paper and report results of the perception task. Accordingly, the “Rel.” metric is computed by dividing the total score by 2000. As for Tables 4/8, both Qwen2.5-VL and InternVL3 report summed scores for perception and cognition in their original settings. Therefore, for these tables, we compute the “Rel.” metric by dividing the score by 2800.
> > >
> > > We hope the above explanations address your concerns, and we welcome further discussions if you have any other questions.

---

### Official Review · Reviewer_5bbK · 2025-06-27

**Clarity:** 3
**Significance:** 3
**Originality:** 3
**Rating:** 5
**Confidence:** 5

**Summary:**

This paper proposes a novel token pruning method to maximize the conditional diversity of retained tokens for efficient VLMs. Based on the conditional similarity between visual tokens conditioned on the instruction, the token pruning problem is reformulated with determinantal point process (DPP). Experimental results demonstrate the effectiveness of the proposed method.

**Questions:**

Overall, I think the proposed method is novel with extensive evaluation. I am concerned about the additional computation cost by extracting textual features and the compatibility with other token pruning methods. If the authors well address my concerns, I will raise the score accordingly.

**Ethical Concerns:**

["NO or VERY MINOR ethics concerns only"]

**Final Justification:**

I think the rebuttal well address my concerns. I raise the score to Accept.

**Quality:**

3

**Strengths And Weaknesses:**

Strengths:
1. The paper is well written and easy to understand. The figure is helpful to understand the whole framework of the proposed method.
2. The proposed conditional similarity is novel and efficient to apply without the need of calculating the attention score.
3. Experiments are extensive with various model variants and, image and video benchmarks, which demonstrate the effectiveness of the proposed method.

Weaknesses:
1. The proposed method needs the additional text embedding for modeling the conditional similarity. Line 145-150, extracting the textual features output by the corresponding text encoders would introduce additional computational cost. I would like to see the analysis of such additional cost.
2. Would the proposed method be combined with token pruning methods within LLMs (such as FastV) for further pruning?

---

> ### Author Rebuttal · Authors · 2025-07-30
>
> We sincerely thank the reviewer for the effort and constructive feedback. We are encouraged by the recognition of our method as *novel and effective*, the acknowledgment of our experiments as *extensive,* and the appreciation of the paper as *well written and easy to understand*. After thoroughly reading the comments, we provide the following point-by-point responses:
>
> 1. **Computational cost of text feature extraction**:
>
>    For text feature extraction, we follow the implementation of TRIM. Since the extraction of visual and textual features is entirely independent, we allocate **separate CUDA streams** for each, allowing both processes to run in parallel. As text feature extraction is generally faster than visual feature extraction, this component **does not introduce additional overhead in practice**. The corresponding pseudocode is as follows:
>    ```python
>    image_stream = torch.cuda.Stream()
>    text_stream = torch.cuda.Stream()
>
>    with torch.cuda.stream(image_stream):
>        image_features = self.vision_tower(images)
>
>    with torch.cuda.stream(text_stream):
>        text_inputs = self.text_tokenizer(texts)
>        text_features = self.text_tower(**text_inputs)
>
>    torch.cuda.synchronize()
>    ```
>
> 2. **Compatibility with token pruning methods within LLM**:
>
>    Our proposed method is fully compatible with any pruning method within LLMs. To demonstrate this, we combine CDPruner with three representative in-LLM pruning methods: FastV, PDrop, and SparseVLM. To retain $m$ visual tokens, we first apply CDPruner before the LLM to reduce the token number to $2m$, and then apply the in-LLM pruning method to further prune it down to $m$ tokens. The experimental results are as follows:
>
>    |Method|VQAV2|GQA|VizWiz|SQA-IMG|TextVQA|POPE|MME|MMB|MMB-CN|MM-Vet|Acc.|Rel.|
>    |:---|:---:|:---:|:---:|:---:|:---:|:---:|:---:|:---:|:---:|:---:|:---:|:---:|
>    |*Upper Bound, All 576 Tokens* **(100%)**|||||||||||||
>    |LLaVA-1.5-7B|78.5|61.9|50.1|69.5|58.2|85.9|1506.5|64.7|58.1|31.3|63.4|100.0%|
>    |*Retain 128 Tokens* **(-77.8%)**|||||||||||||
>    |CDPruner|76.6|59.9|52.8|69.0|56.2|87.7|1431.4|63.1|55.0|32.8|**62.5**|**99.0%**|
>    |CDPruner+FastV|76.4|58.2|51.3|68.3|57.4|81.8|1469.6|62.8|56.3|32.3|61.8|98.1%|
>    |CDPruner+PDrop|76.9|59.6|50.9|67.9|57.4|85.2|1473.9|63.6|55.9|32.3|62.3|98.7%|
>    |CDPruner+SparseVLM|76.8|59.5|51.3|67.7|57.4|86.1|1440.4|63.1|56.3|32.6|62.3|98.7%|
>    |*Retain 64 Tokens* **(-88.9%)**|||||||||||||
>    |CDPruner|75.4|58.6|53.4|68.1|55.3|87.5|1415.1|61.1|53.2|30.5|61.4|97.0%|
>    |CDPruner+FastV|75.3|57.4|52.6|69.6|56.6|82.2|1417.1|62.4|53.1|30.3|61.0|96.5%|
>    |CDPruner+PDrop|74.3|57.9|52.3|69.9|56.1|84.9|1406.6|62.2|54.2|30.0|61.2|96.7%|
>    |CDPruner+SparseVLM|75.7|58.3|52.4|69.5|55.9|86.3|1434.4|61.8|54.3|30.6|**61.7**|**97.4%**|
>    |*Retain 32 Tokens* **(-94.4%)**|||||||||||||
>    |CDPruner|73.6|57.0|53.1|69.5|53.2|87.9|1373.0|59.6|49.6|27.8|60.0|94.3%|
>    |CDPruner+FastV|73.4|56.1|52.9|68.7|54.8|81.1|1368.9|60.0|50.9|28.3|59.5|93.9%|
>    |CDPruner+PDrop|73.5|56.5|52.5|69.8|54.4|83.2|1366.1|60.1|51.2|28.9|59.8|94.4%|
>    |CDPruner+SparseVLM|74.1|56.9|52.8|69.3|54.5|85.5|1392.7|60.1|52.4|29.3|**60.5**|**95.4%**|
>
>    These combined methods yield further performance improvements over the original CDPruner, clearly demonstrating that our approach is fully compatible with various in-LLM pruning approaches.
>
> In summary, through the implementation of dual CUDA streams, **CDPruner does not introduce additional computational cost for text feature extraction**, confirming its efficiency. Meanwhile, experimental results show that **CDPruner is also compatible with a wide range of pruning methods within LLMs** and achieve improvements, fully demonstrating its compatibility. We hope the above response addresses your concerns.

---

> > ### Comment · Reviewer_5bbK · 2025-08-03
> >
> > Thanks for the rebuttal. I think the paper is above the threshold and I will raise the score to Accept. Besides, I am interested in evaluating the proposed method based on Qwen2.5-VL model. Could the authors provide the source code for evaluating the proposed method?

---

> > > ### Author Response · Authors · 2025-08-04
> > >
> > > We thank the reviewer for the positive response and are pleased that our rebuttal has addressed your concerns. Due to the NeurIPS policy of this year, authors are not allowed to **contain any links to external pages** in the responses at both rebuttal and discussion phases. Therefore, we are unable to directly share the implementation code of our method on Qwen2.5-VL at this stage. The authors promise to open source this part of the code after the submission process. Once again, we sincerely appreciate the time and effort you have dedicated to reviewing our work.

---

### Official Review · Reviewer_pdN1 · 2025-06-27

**Clarity:** 3
**Significance:** 3
**Originality:** 4
**Rating:** 5
**Confidence:** 4

**Summary:**

The paper proposes CDPruner, a novel visual token pruning method for multimodal large language models (MLLMs) to reduce inference cost. It defines conditional similarity between visual tokens based on instructions and reformulates the token pruning problem using determinantal point process (DPP) to maximize conditional diversity. CDPruner is training-free and model-agnostic, achieving state-of-the-art performance on various vision-language benchmarks while significantly reducing computational cost.

**Questions:**

- The paper states that CDPruner is model-agnostic. Could the authors provide further information on potential challenges and corresponding strategies when adapting it to different types of visual encoders and language models?
- The effectiveness of the CDPruner method is significant. Could the authors provide an in-depth analysis of the reasons behind CDPruner’s effectiveness? Such analysis would greatly benefit research in this field.

If the authors address these issues, I will further increase my score.

**Ethical Concerns:**

["NO or VERY MINOR ethics concerns only"]

**Final Justification:**

After reading the authors' rebuttal, I find that my main concerns have been satisfactorily addressed.
The authors clarified how CDPruner maintains generalizability across different embedding models, alleviating my concern about its applicability beyond specific architectures.
While multi-turn dialogue was not a focus of the current work, the authors provided reasonable justification and future directions for extending CDPruner to such scenarios.
The explanation regarding token-level contributions and pruning rationale was helpful in clarifying the interpretability aspect of the method.
Given these responses, I now have greater confidence in the soundness and potential of the proposed approach. I will raise my score accordingly. I encourage the authors to further explore generalization and multi-turn capabilities in future work.

**Limitations:**

The author discusses the limitations in supplementary materials.

**Paper Formatting Concerns:**

The manuscript generally adheres to the NeurIPS formatting requirements. However, there is a minor issue: the font size in certain figures and tables is slightly small, which may affect readability when printed.

**Quality:**

4

**Strengths And Weaknesses:**

- Strengths:
  - This paper proposes a novel method, CDPruner, which surpasses traditional attention-based or similarity-based token pruning approaches. By defining conditional similarity and leveraging Determinantal Point Processes (DPP) to maximize conditional diversity, it offers a unique perspective for selecting diverse and relevant visual tokens. This represents a significant advancement in improving the inference efficiency of MLLMs.
  - CDPruner is model-agnostic and requires no additional training, granting it high flexibility and allowing easy application to various MLLMs without further training. This feature saves time and computational resources, facilitating seamless integration into different models and architectures.
  - The authors conduct a comprehensive evaluation of CDPruner across multiple image and video understanding tasks, covering various benchmark datasets. This thorough assessment provides strong evidence of the method’s robustness and generalizability in different scenarios and datasets.

- Weaknesses:
  - The visual and textual embeddings vary across different models, which may affect the applicability of the CDPruner method. It may be beneficial to consider optimizing the generalizability of CDPruner from the perspectives of visual and textual embeddings, such as by introducing additional embedding models.
  - Although this is understandable, the paper does not consider applications in multi-turn dialogue scenarios. Is it possible to dynamically adjust the pruned tokens in multi-turn contexts? I suggest investigating the extent to which pruning affects the performance of multi-turn dialogues.
  - Although CDPruner achieves excellent performance, further exploration of its underlying principles, such as the semantic contribution of key tokens, is warranted.

---

> ### Author Rebuttal · Authors · 2025-07-30
>
> We thank the reviewer for the detailed review and valuable feedback. We are encouraged by the recognition of our method as *novel and offering a unique perspective*, as well as the acknowledgment that our evaluation is *comprehensive and thorough*. After careful consideration of the comments, we provide the following point-by-point responses:
> 1. **Additional embedding models for instruction relevance**:
>
>    To compute the relevance between visual tokens and user instructions, it is necessary to obtain visual and textual embeddings of the same dimension and ensure they are situated in a unified semantic space. Therefore, the most suitable choice is a model that has been pre-trained through language-image contrastive pretraining. If the visual features in the MLLM are already well aligned with text, an additional embedding model may not be required. Here, we introduce the CLIP model for Qwen2.5-VL to validate the effectiveness of an additional embedding model. The corresponding experimental results are as follows:
>
>    |Method|AI2D|HallBench|MMB-EN|MMB-CN|Acc.|Rel.|
>    |:---|:---:|:---:|:---:|:---:|:---:|:---:|
>    |*Upper Bound, All 1296 Tokens* **(100%)**|||||||
>    |Qwen2.5-VL-7B|84.4|46.8|83.9|83.4|74.6|100.0%|
>    |*Retain 512 Tokens* **(-60.5%)**|||||||
>    |CDPruner|82.9|42.5|82.2|82.6|72.6|96.5%|
>    |CDPruner+CLIP|**83.7**|**43.6**|**83.1**|**82.1**|**73.1**|**97.5%**|
>    |*Retain 256 Tokens* **(-80.2%)**|||||||
>    |CDPruner|80.5|40.1|80.9|79.9|70.4|93.3%|
>    |CDPruner+CLIP|**81.7**|**41.5**|**82.6**|**81.1**|**71.7**|**95.3%**|
>    |*Retain 128 Tokens* **(-90.1%)**|||||||
>    |CDPruner|76.0|37.2|76.2|76.5|66.5|88.0%|
>    |CDPruner+CLIP|**79.3**|**38.1**|**81.3**|**78.1**|**69.2**|**91.5%**|
>
>    After incorporating CLIP into Qwen2.5-VL, CDPruner consistently achieves improved performance across various benchmarks. This result indicates that accurate instruction relevance estimation plays a crucial role in effective token pruning and provides new insights for adapting CDPruner to different MLLM architectures. We will continue to explore the value of additional embedding models for token pruning in our future work.
>
> 2. **Extension to multi-turn dialogues**:
>
>    In multi-turn scenarios, it is possible to dynamically adjust the pruned tokens based on the current instruction by caching the key-value pairs of all visual tokens. Specifically, during the prefill stage, instead of discarding all pruned tokens, we can encode and store their key and value representations. In subsequent dialogue turns, we can then retrieve relevant tokens from the cache based on the current instruction, enabling dynamic token adjustment. Following the reviewer’s suggestion, we directly apply CDPruner to the MMDU benchmark [1] without performing dynamic token adjustment to evaluate the effect of pruning under this challenging setting:
>
>    |Method|C|R|VP|LC|AA|IRU|Acc.|Rel.|
>    |:---|:---:|:---:|:---:|:---:|:---:|:---:|:---:|:---:|
>    |*Upper Bound, All 576 Tokens* **(100%)**|||||||||
>    |LLaVA-1.5-7B|34.8|32.7|39.4|65.3|47.4|39.5|42.9|100.0%|
>    |*Retain 128 Tokens* **(-77.8%)**|||||||||
>    |TRIM|35.7|34.2|38.7|64.6|46.8|39.2|42.8|99.8%|
>    |CDPruner|**36.2**|**34.9**|**40.0**|**66.2**|**48.0**|**40.8**|**44.0**|**102.6%**|
>    |*Retain 64 Tokens* **(-88.9%)**|||||||||
>    |TRIM|35.6|34.1|37.1|63.8|44.8|37.7|41.7|97.2%|
>    |CDPruner|**36.1**|**34.4**|**38.6**|**64.5**|**46.2**|**39.0**|**42.8**|**99.8%**|
>    |*Retain 32 Tokens* **(-94.4%)**|||||||||
>    |TRIM|35.4|34.0|36.1|62.8|44.2|36.9|41.2|96.0%|
>    |CDPruner|**35.6**|**34.0**|**36.7**|**62.9**|**44.6**|**38.0**|**41.5**|**96.7%**|
>
>    TRIM prunes tokens based on their relevance to the instruction, which is suboptimal for multi-turn dialogue. If the subsequent question differs significantly from the previous one, the retained tokens may no longer be relevant, resulting in degraded performance. In contrast, CDPruner incorporates diversity modeling via DPP, which enables it to preserve more informative and comprehensive visual content while still considering relevance. As a result, it maintains better performance even in multi-turn scenarios. Experimental results show that CDPruner consistently outperforms TRIM under different reduction ratios, demonstrating its adaptability to multi-turn dialogues.
>
> 3. **In-depth analysis of CDPruner’s effectiveness**:
>
>    This work identifies visual diversity and instruction relevance as two key factors for token pruning in MLLMs. Previous methods typically considered only one of these factors. Methods focusing solely on diversity (e.g., DART, DivPrune) tend to retain too many tokens irrelevant to the current question, while a large number of tokens in key areas are discarded, leading to loss of critical information. On the other hand, methods focusing only on relevance (e.g., FastV, TRIM) often retain redundant tokens while discarding informative ones from low-relevance regions, also resulting in performance degradation. Naively combining both factors in a two-stage pruning strategy does not effectively address this issue, as each stage may still retain unimportant tokens. In contrast, CDPruner jointly models both relevance and diversity using DPP, and selects the optimal subset from a global perspective. This enables it to retain the most relevant tokens while removing redundancy, achieving efficient pruning without sacrificing performance.
>
> 4. **Potential challenges across different architectures and corresponding strategies**:
>
>    Since CDPruner performs token pruning before the language model, the visual encoder is more crucial to our method. For MLLMs whose visual encoders are paired with a corresponding text encoder (e.g., LLaVA-1.5, LLaVA-OneVision), we directly compute cosine similarity between visual and textual embeddings for relevance estimation. For models without a paired text encoder (e.g., Qwen2.5-VL, InternVL3), alternative strategies can be adopted, such as using embeddings from the language model’s embedding layer to compute similarity, leveraging text-visual attention from shallow layers of the language model, or introducing an additional embedding model (e.g., CLIP) to assist relevance computation. Designing a general pruning strategy for diverse MLLM architectures remains a promising direction, and we will continue to explore it in future work.
>
> 5. **Font size issue**:
>
>    Thanks for pointing this out. We will increase the font size in the corresponding figures and tables in the revised version to improve the readability of the paper when printed.
>
> We sincerely appreciate the reviewer’s recognition of our work. And we hope that the above analysis of CDPruner’s effectiveness and the discussion on adaptation strategies to various model architectures have addressed your concerns and can offer valuable insights for future research in this field.
>
> &nbsp;
>
> [1] Liu, Ziyu, et al. "Mmdu: A multi-turn multi-image dialog understanding benchmark and instruction-tuning dataset for lvlms." Advances in Neural Information Processing Systems 37 (2024): 8698-8733.

---

> > ### Comment · Reviewer_pdN1 · 2025-08-04
> >
> > Thank you for the detailed rebuttal. My main concerns have been adequately addressed, and I will raise my score accordingly. As a follow-up, could you please list all the model series that have successfully adopted the proposed method so far?

---

> > > ### Author Response · Authors · 2025-08-04
> > >
> > > We thank the reviewer for the positive comments and are pleased that our response has addressed your concerns. Regarding model compatibility, we have successfully applied CDPruner to a range of models, including the LLaVA series (LLaVA-1.5, LLaVA-NeXT, LLaVA-OneVision, and LLaVA-Video), the Qwen-VL series (Qwen2.5-VL), and the InternVL series (InternVL3). The complete implementation for all supported models will be open-sourced after the submission process concludes. We also plan to extend CDPruner to more MLLMs in future work. Once again, we sincerely appreciate the time and effort you have dedicated to reviewing our work.

---

### Official Review · Reviewer_w7Zd · 2025-07-04

**Clarity:** 3
**Significance:** 3
**Originality:** 3
**Rating:** 4
**Confidence:** 4

**Summary:**

This paper tackles a critical and practical problem in MLLMs: the high inference cost caused by a large number of visual tokens. The authors introduce CDPruner, a novel, training-free pruning method that intelligently reduces visual tokens by maximizing the "conditional diversity" of the retained set. The paper is well-motivated, the proposed method is elegant, and the experiments are comprehensive and impressive.

**Questions:**

N/A

**Ethical Concerns:**

["NO or VERY MINOR ethics concerns only"]

**Quality:**

3

**Strengths And Weaknesses:**

**Strength**:

The work astutely identifies the core trade-off in token pruning (relevance vs. diversity) and proposes an elegant solution. The concept of "conditional diversity," modeled via a Determinantal Point Process (DPP), is a novel and powerful way to unify these competing goals, providing a fresh perspective that moves beyond simpler heuristics.

The authors validate CDPruner across diverse MLLM architectures and challenging tasks (including high-resolution and video inputs). The results are compelling, demonstrating remarkable performance retention (e.g., >94% accuracy with only 5.6% of tokens) and significant efficiency gains in latency and FLOPs. The breadth of experiments strongly supports the method's effectiveness.


**Weakness**:

While the paper claims the additional overhead of CDPruner is "negligible" (<10ms), a more detailed analysis would be beneficial. The paper states the complexity is O(nm²), but it would be helpful to see how this overhead scales in practice with the number of initial tokens n and kept tokens m. In scenarios with fewer visual tokens or in real-time applications with extremely strict latency requirements, this "fixed cost" might become a non-trivial factor. A more granular analysis would help readers better assess the cost-benefit trade-off in different settings.

---

> ### Author Rebuttal · Authors · 2025-07-30
>
> We sincerely thank the reviewer for the time and effort in evaluating our work. We are encouraged by the recognition of our method as *elegant* and our experiments as *comprehensive and impressive*. After carefully reading the comments, we would like to address the reviewer’s concerns regarding the computational overhead of DPP in our CDPruner as follows:
> - **Computational overhead with respect to the initial token number $n$**:
>
>   We fix the number of kept tokens to 64 and measure the runtime of a single DPP step (averaged over 10,000 runs) as a function of the initial token number $n$:
>
>   |Initial Token Number ($n$)|144|288|576|1152|2304|
>   |:---|:---:|:---:|:---:|:---:|:---:|
>   |Latency (ms)|0.86|1.09|1.37|2.00|3.64|
>
>   The results show that the overall computational overhead does not increase significantly with the initial token number $n$. Even when the number of visual tokens exceeds 2k, the latency of a single run remains below 4ms, which is well within an acceptable range for practical applications.
>
> - **Computational overhead with respect to the kept token number $m$**:
>
>   We fixed the number of initial tokens to 576 and measured the runtime of a single DPP step (averaged over 10,000 runs) as a function of the kept token number $m$:
>
>   |Kept Token Number ($m$)|16|32|64|128|256|
>   |:---|:---:|:---:|:---:|:---:|:---:|
>   |Latency (ms)|0.45|0.77|1.37|2.73|5.84|
>
>   The results show that while the runtime increases more noticeably with $m$ compared to $n$, the latency per run still remains very low. In practical scenarios, the number of kept tokens $m$ is typically much smaller than the initial number $n$, so the overall DPP runtime remains negligible and does not affect the real-time application of the algorithm.
>
> - **Further optimization of algorithmic complexity**:
>
>   Our current implementation adopts fast greedy MAP inference for DPP, with a theoretical complexity of $\mathcal{O}(nm^2)$. However, for MLLMs with dynamic high-resolution inputs (e.g., LLaVA-NeXT, InternVL3), the input image is first divided into equally sized patches, and the DPP can be performed in parallel across different patches. This significantly reduces the computational complexity: for an image with $k$ patches, the global complexity $\mathcal{O}((kn)(km)^2) = \mathcal{O}(k^3nm^2)$ can be reduced to a local complexity of $\mathcal{O}(knm^2)$.
>
>   Additionally, by applying a sliding window of width $w (w < m)$ during DPP computation, the complexity can be further reduced to $\mathcal{O}(wnm)$ [1]. Furthermore, incorporating a Markov chain approximation into the DPP sampling can reduce the complexity to the ideal $\mathcal{O}(m^3)$ [2].
>
> - **Further reduction of runtime latency**:
>
>   Given that the overall computing process of DPP is sequential, in real-time applications with extremely strict latency requirements, we can further optimize inference latency by implementing custom operators, enabling low-level and system-level acceleration.
>
> In summary, we believe the computational overhead of DPP does not hinder the real-time applicability of CDPruner. Moreover, various strategies exist to further reduce the inference latency under different deployment scenarios. We hope the above responses effectively address your concerns, and we sincerely hope you would consider increasing the rating if your concerns have been resolved.
>
> &nbsp;
>
> [1] Chen, Laming, Guoxin Zhang, and Eric Zhou. "Fast greedy map inference for determinantal point process to improve recommendation diversity." Advances in neural information processing systems 31 (2018).
>
> [2] Kang, Byungkon. "Fast determinantal point process sampling with application to clustering." Advances in Neural Information Processing Systems 26 (2013).

---

### Comment · Area_Chair_Kjgf · 2025-08-04
**Engage in Author-Reviewer Discussions**

Dear reviewers,

If you haven't done so already, please click the 'Mandatory Acknowledgement' button and actively participate in the rebuttal discussion with the authors after carefully reading all other reviews and the author responses.

Thanks,
AC

---

### Note · Authors · 2025-08-15

We sincerely thank the ACs, SACs, PCs, and all reviewers for their valuable time and constructive feedback throughout the review process. The high-quality rebuttal stage greatly helped address reviewers’ concerns and further improved the quality of our work. Here we sequentially present the strengths highlighted by reviewers, and their concerns together with our responses as the final remarks.

### **Strengths Highlighted by Reviewers**
We are greatly encouraged that **all reviewers provided positive initial ratings**. All reviewers recognized the novelty of our method and the thoroughness of our evaluation. Reviewer w7Zd acknowledged that our work is well-motivated and provides an elegant solution. Reviewer pdN1 recognized that it offers a unique perspective and is highly flexible (model-agnostic). Reviewer 5bbK commended the paper for being well written and easy to understand, while Reviewer 4nPZ affirmed that its performance is consistently superior.

### **Reviewers’ concerns and our responses**
During the rebuttal stage, we fully addressed all reviewers’ concerns.

1. Additional Computational Overhead

Reviewer w7Zd expressed concerns regarding the overhead introduced by DPP. We conducted an in-depth analysis of the computational cost under various settings and provided suggestions for further optimization. Reviewer 5bbK was concerned about the overhead of extracting text features; we clarified that this cost can be completely eliminated via separate CUDA streams.

2. Generality and Compatibility

Reviewer pdN1 questioned the generalizability of our method across different models. We incorporated CLIP as an additional embedding model, which further improved performance. Reviewer 5bbK raised concerns about compatibility with other in-LLM pruning algorithms; we demonstrated that combining CDPruner with methods such as FastV yielded further performance gains.

3. Extension to Multi-Turn Dialogues

Reviewer pdN1 questioned the applicability of our method to multi-turn dialogue scenarios. We extended CDPruner to handle such settings, evaluated it on the MMDU benchmark, and achieved outstanding performance.

4. Details of Evaluation Metrics

Reviewer 4nPZ expressed uncertainty regarding the metrics used in our evaluation. We supplemented additional details and released the implementation code for metric computation, fully resolving the concern.

As a result of our responses, **all reviewers indicated that their concerns had been adequately addressed**.

---

### Decision · Program_Chairs · 2025-09-17

**Decision:**

Accept (poster)

**Comment:**

This paper proposes a novel visual token pruning method, called CDPruner, for multimodal LLMs based on determinantal point process (DPP). In specific, CDPruner tries to maximize both of the diversity between visual tokens and the relevance to input text tokens, especially without any additional training. Experimental results on various image and video understanding benchmarks with LLaVA series and Qwen2.5-VL show that CDPruner maintains performances while significantly reducing visual tokens and corresponding computational cost.

Overall, the proposed CDPruner seems to be technically sound and novel with clear motivation. Moreover, rigorous empirical validation supports its feasibility and powerfulness. Most concerns raised by the reviewers are well addressed by the authors, including the additional runtime for pruning. Based on the consensus among the reviewers, I would recommend this paper to be accepted.

From the perspective of optimal pruning, I think it would be beneficial to reflect on the dependency of the MLLM backbone model and its underlying reasoning process. In addition, it would strengthen the work to experimentally evaluate whether further finetuning of the MLLM with the proposed pruning method leads to additional performance improvements.